



# Marine boundary layer cloud property retrievals from high–resolution ASTER observations: Case studies and comparison with Terra–MODIS

Frank Werner[1], Galina Wind[2], Zhibo Zhang[3], Steven Platnick[2], Larry Di Girolamo[4], Guangyu Zhao[4], Nandana Amarasinghe[5], and Kerry Meyer[2,6]

[1]Joint Center for Earth Systems Technology, 5523 Research Park Drive, Baltimore, MD 21228, USA
[2]NASA Goddard Space Flight Center, Greenbelt, Maryland, 20771, USA
[3]Physics Department, University of Maryland, Baltimore County, 1000 Hilltop Circle, Baltimore MD 21228, USA
[4]Department of Atmospheric Sciences, University of Illinois at Urbana–Champaign, 105 South Gregory Street, Urbana, IL 61801, USA
[5]Science Systems and Applications, Inc., NASA/GSFC, Greenbelt, MD 20771, USA
[6]Goddard Earth Sciences Technology and Research (GESTAR), Universities Space Research Association, Columbia, Maryland, 21046, USA

*Correspondence to:* F. Werner (frankw@umbc.edu)

**Abstract.** Retrievals of marine boundary layer (MBL) cloud microphysical and optical properties, based on high spatial resolution observations from the Advanced Spaceborne Thermal Emission and Reflection Radiometer (ASTER), are presented. The research–level retrieval algorithm, which is modified from the MODIS Data Collection 6 (C6) operational algorithms, is documented.

An ASTER specific cloud masking scheme is implemented and the derived cloud covers agree well with those from an independent scheme reported in a previous ASTER cloud cover study by (Zhao and Di Girolamo, 2006). The viability of the retrieval algorithm is shown by a comparison of retrieved cloud optical thickness $\tau_{\mathrm{AaM}}$ and effective droplet radius $r_{\mathrm{eff,AaM}}$ with the co–located, operational MODIS retrievals for 48 MBL cloud scenes with different degrees of cloud heterogeneity.

The input for the retrieval is provided by aggregated, atmospherically corrected ASTER cloud top reflectances $\widehat{\gamma}_{0.86,\mathrm{AaM}}$ and $\widehat{\gamma}_{2.1,\mathrm{AaM}}$. By means of the subpixel cloud cover $C_{\mathrm{sub}}$, which is derived from the ASTER cloud mask at its native resolution, the data set is divided into $52,254$ overcast and $47,538$ partially cloudy pixels. PDFs and scatter plots of $\widehat{\gamma}_{0.86,\mathrm{AaM}}$, $\widehat{\gamma}_{2.1,\mathrm{AaM}}$, and $\tau_{\mathrm{AaM}}$ agree well with the corresponding MODIS products with Pearson's product–moment correlation coefficients of $R > 0.980$. A comparison for pixels with $C_{\mathrm{sub}} = 1.0$ yields a high agreement between $r_{\mathrm{eff,AaM}}$ and the MODIS retrievals with $R = 0.972$. However, for partially cloudy pixels there are significant differences between both products which can exceed $10\,\mu\mathrm{m}$. Moreover, it is shown that the numerous delicate cloud structures in the example MBL scenes, resolved by the high–resolution ASTER retrievals, are smoothed by the MODIS observations. The overall good agreement between





the research–level ASTER results and the operational MODIS C6 products proves the feasibility
of MODIS–like retrievals from ASTER reflectance measurements and provides the basis for fu-
ture studies concerning the scale–dependency of satellite observations and 3–dimensional radiative
effects.

## 1 Introduction

Marine boundary layer (MBL) clouds are an important component of the Earth's climate system.
Compared to the underlying ocean surface they are characterized by an increased albedo in the
visible spectral wavelength range, while the radiation emitted in the thermal infrared spectral wave-
length range is largely the same (Warren et al., 1988; Albrecht et al., 1995). This means that MBL
scenes exhibit a negative radiative forcing at the Earth's top–of–atmosphere (Ramanathan et al.,
1989; Harrison et al., 1990; Klein and Hartmann, 1993). MBL clouds are especially sensitive to
aerosol indirect effects such as the cloud albedo effect (Twomey, 1977; Ackerman et al., 2000;
McFarquhar et al., 2004; Werner et al., 2014) and the cloud lifetime effect (Albrecht, 1989; Lohmann and Feichter,
2005).

The radiative effects of MBL clouds on the climate system are largely determined by cloud frac-
tion, cloud optical thickness $\tau$ and effective droplet radius $r_{\mathrm{eff}}$. Among others, the Moderate Resolu-
tion Imaging Spectroradiometer (MODIS) provides global retrievals of these cloud variables. How-
ever, derived cloud properties from passive remote sensing techniques are inherently dependent on
the spatial resolution of the observations and thus the influence of cloud horizontal heterogeneities.
Studies on scale–dependent uncertainties in estimated cloud amount due to unresolved clear–sky
contaminations have been reported by Shenk and Salomonson (1972), Wielicki and Parker (1992),
DiGirolamo and Davies (1997) and Dey et al. (2008). This dependence on spatial resolution also
extents to the retrievals of $\tau$ and $r_{\mathrm{eff}}$, which usually are achieved with the help of the bispectral
solar reflective method (Twomey and Seton, 1980; Nakajima and King, 1990; Rossow and Schiffer,
1991) and under the independent pixel approximation (IPA, see Cahalan et al., 1994a, b).

The IPA introduces two general three–dimensional (3D) radiative effects, caused by applying 1–
dimensional (1D) radiative transfer on 3D cloud structures. For observations with a high spatial
resolution, such as the Thematic Mapper onboard the Landsat satellites, cloud heterogeneities at
scales larger than the sensor spatial resolution yield a breakdown of IPA (Barker and Liu, 1995;
Chambers et al., 1997). In contrast, satellite observations with a lower spatial resolution cannot re-
solve heterogeneous cloud structures within a pixel, introducing significant biases in retrieved $\tau$
and $r_{\mathrm{eff}}$ (Cahalan et al., 1994a; Marshak et al., 2006; Zhang and Platnick, 2011; Zhang et al., 2012).
Depending on the spatial resolution of the observations, the biases due to resolved or unresolved vari-
ability can vary in magnitude and even compensate each other to a certain degree (Marshak et al.,
2006).



The increase in sensor spatial resolution of space–borne instruments, while introducing increased detail in retrieved cloud parameters even for heterogeneous cloud fields, compounds the importance of 3D radiative transfer in the cloudy atmosphere. The bispectral retrieval approach is adopted by several major satellite missions, most prominently by the MODIS instrument. MODIS provides operational cloud products sampled on a global scale with a horizontal resolution of $1000\,\mathrm{m}$, which makes
the retrieved MODIS cloud variables susceptible to biases introduced by resolved and unresolved variability. Studies on satellite observations of unresolved cloud variability require a significantly higher spatial resolution. Samples by the Advanced Spaceborne Thermal Emission and Reflection Radiometer (ASTER) are characterized by a horizontal resolution as high as $15\,\mathrm{m}$ (Abrams, 2000). While ASTER data are usually applied to study changes in land cover and biophysical parameters
(Stefanov and Netzband, 2005), there are a number of studies deploying ASTER for cloud observations. Zhao and Di Girolamo (2006, 2007) and Dey et al. (2008) use the high-resolution ASTER reflectance measurements at the $\lambda = 0.86\,\mu\mathrm{m}$ band to derive a statistical description of the macrophysical properties of trade wind clouds. Seiz et al. (2006) and Genkova et al. (2007) employ the stereoscopic capabilities of ASTER, as well as its window–IR channel, for studying cloud top heights.
Jones et al. (2012) use the high–resolution ASTER data as a training set for a pattern recognition approach for a new algorithm implemented by the MISR team to improve cloud fraction estimates. Finally, Marshak et al. (2006) and Wen et al. (2007) report on MODIS cloud property retrievals and apply ASTER reflectances to gain a better understanding of the cloud 3D structure on the MODIS microphysical cloud retrievals. Despite all these studies, there remains no cloud property retrieval
algorithm for ASTER.

In this study MODIS–like retrievals of $\tau$ and $r_{\mathrm{eff}}$ based on high–resolution ASTER observations are presented. Both ASTER and MODIS are aboard NASA's Terra satellite, which allows for inter–comparison studies and cross validation of the retrieval products. The objectives of this study are as follows: i) Documenting the research–level retrieval setup which provides cloud property retrievals
based on ASTER observations, and ii) comparing co–located ASTER retrievals with the operational MODIS C6 results for 48 MBL scenes with different degrees of horizontal heterogeneity. It is shown that estimates of $\tau$ and $r_{\mathrm{eff}}$ from ASTER measurements are consistent with the operational MODIS data products. Thus, the combination of high–resolution ASTER observations and the presented retrieval setup provides a unique framework for future studies on the reliability of retrievals for
partially cloudy pixels, the scale dependence of satellite–based remote sensing products and the influence of 3D radiative effects.

## 2   ASTER and MODIS

Data in this study are provided by ASTER, as well as MODIS. This Section provides an introduction to the ASTER instrument and a detailed description of the steps necessary to obtain reflectances from





the raw ASTER observations. A brief overview of MODIS is also given. Differences between the
spectral response functions of both instruments are presented.

### 2.1   ASTER

ASTER is an imaging spectroradiometer installed aboard the multi–national scientific research satel-
lite Terra (EOS AM–1). Information on the instrument design and science objectives can be found in
Yamaguchi and Hiroji (1993); Yamaguchi et al. (1998) and Abrams (2000). ASTER collects data in
fifteen distinct spectral bands, covering the visible to the thermal infrared spectral wavelength range.
The spatial resolution of an individual ASTER pixel in the visible to near–infrared spectral wave-
length range (VNIR) is $15\,\mathrm{m}$, while it is $30\,\mathrm{m}$ and $90\,\mathrm{m}$ in the in the shortwave–infrared (SWIR) and
thermal infrared (IR) spectral wavelength range, respectively. Table 1 lists the spectral band num-
bers and the respective wavelength ranges $\Delta\lambda$ that the ASTER instrument covers. While all bands
are operated in nadir–viewing mode, the $\lambda = (0.760 - 0.860)\,\mu\mathrm{m}$ band also provides a backward–
viewing direction. From the $10:30$ AM sun–synchronous orbit of Terra, ASTER samples roughly
650 scenes daily. Each of these scenes covers an area of $(60 \cdot 60)\,\mathrm{km}^2$. However, ASTER data sam-
pled over ocean surfaces are usually not archived and observations of MBL clouds are provided by
specific scientific objectives, as reviewed in Jones et al. (2012).

The digital ASTER counts $d_\mathrm{A}(\Delta\lambda)$ that are sampled by the instrument over a cloud scene are con-
verted into spectral ASTER radiances $I_\mathrm{A}(\Delta\lambda)$ via the conversion equation provided by Abrams et al.
(2004):

$$I_\mathrm{A}(\Delta\lambda) = \big(d_\mathrm{A}(\Delta\lambda) - 1\big) \cdot C_\mathrm{N}. \qquad (1)$$

The unit conversion coefficient $C_\mathrm{N}$ for each band, which is dependent on the respective gain setting
provided in the embedded metadata of each ASTER data container, is also given in Table 1. Spectral
ASTER reflectances $\gamma_\mathrm{A}(\Delta\lambda, \theta_0)$ are calculated by:

$$\gamma_\mathrm{A}(\Delta\lambda, \theta_0) = \frac{\pi \cdot I_\mathrm{A}(\Delta\lambda) \cdot r_\mathrm{SE}^2}{F_0(\Delta\lambda) \cdot \cos\theta_0}, \qquad (2)$$

where $r_\mathrm{SE}$ denotes the distance between the Earth and the Sun in astronomical units and $F_0(\Delta\lambda)$ is
the incoming spectral solar irradiance modified by the solar zenith angle $\theta_0$. Band–specific $F_0(\Delta\lambda)$
values are obtained from the convolution of high resolution spectral solar irradiances with the
respective spectral response function (SRF) of each ASTER band. The high resolution spectral
solar irradiance between $\lambda = (0.199 - 0.539)\,\mu\mathrm{m}$ is provided by Thuillier et al. (2003), between
$\lambda = (0.540 - 1.060)\,\mu\mathrm{m}$ by Neckel and Labs (1984), and between $\lambda = (1.450 - 400.000)\,\mu\mathrm{m}$ by
Thekaekara (1974). The specific ASTER response functions are provided by the ASTER science
team on the instrument website (http://asterweb.jpl.nasa.gov/characteristics.asp).

Absolute radiometric uncertainties $\delta$ of VNIR and SWIR reflectances are $< 4\,\%$ (Yamaguchi et al.,
1998). However, due to an increase in the SWIR detector temperature starting in May 2007, the





SWIR signal started to suffer from anomalous striping and saturation of values. While the VNIR and
IR bands are not affected, no reliable SWIR data sampled after that date are available (with brief
exceptions in June and July 2007, as well as January to April 2008).

## 2.2 MODIS

The scanning radiometer MODIS is installed aboard NASA's Terra and Aqua (EOS PM–1) plat-
forms, launched in 1999 and 2002, respectively. MODIS has a viewing swath width of $2330\,\mathrm{km}$. To-
gether with the orbit characteristics of the Terra (and Aqua) platform this allows for a global coverage
every two days. MODIS collects data in 36 spectral bands between $(0.415 - 14.235)\,\mu\mathrm{m}$. Except for
a number of bands, the general spatial resolution of a MODIS pixel is $1000\,\mathrm{m}$. Further information
on MODIS and the cloud product algorithms is given in Ardanuy et al. (1992), Barnes et al. (1998),
and Platnick et al. (2003).

The current version of the cloud product algorithm, and the one that yields the data in this study, is
Data Collection 6 (C6). This new set of algorithms includes a flag for partially cloudy (PCL) pixels.

## 2.3 Comparison of Spectral Response Functions

The MODIS cloud property retrieval is based on reflectances sampled in two spectral bands, one in
the VNIR and one in the SWIR. Although ASTER employs similar bands in these spectral regions,
differences in the respective SRF can impact the retrieval. It is therefore important to understand the
behavior of the ASTER SRFs and the respective band deviations from the MODIS instrument. For
MODIS the VNIR reflectance $\gamma_{0.86,\mathrm{M}}$ is provided by band 2, which covers $\lambda = (0.841 - 0.876)\,\mu\mathrm{m}$
and is centered around $\lambda = 0.8585\,\mu\mathrm{m}$, while the SWIR reflectances $\gamma_{2.1,\mathrm{M}}$ are sampled by band
7, which covers $\lambda = (2.105 - 2.155)\,\mu\mathrm{m}$ and is centered around $\lambda = 2.130\,\mu\mathrm{m}$. VNIR and SWIR
reflectances $\gamma_{0.86,\mathrm{A}}$ and $\gamma_{2.1,\mathrm{A}}$ for ASTER are detected at bands 3N (nadir–viewing mode) and 5,
respectively.

Figure 1(a) shows the SRF of the ASTER (black) and MODIS (green) VNIR band as a function of
wavelength $\lambda$. Compared to MODIS, the SRF of the ASTER VNIR band is significantly broader with
a spectral width of about $\Delta\lambda = 0.100\,\mu\mathrm{m}$ (compared to $\Delta\lambda = 0.060\,\mu\mathrm{m}$ for MODIS). Moreover, the
center of the SRF is shifted by about $\lambda = 0.050\,\mu\mathrm{m}$ towards smaller wavelengths.

Figure 1(b) shows the SRF of the applied ASTER and MODIS SWIR bands, respectively. Com-
pared to MODIS, the center of the ASTER SRF is shifted by about $\lambda = 0.035\,\mu\mathrm{m}$ towards larger
wavelengths and the spectral width is decreased by about $\Delta\lambda = 0.004\,\mu\mathrm{m}$.

Implications of the SRF differences on the cloud property retrieval are discussed in Section 3.3.





## 3 Cloud Property Retrieval Algorithm

In this section an ASTER–specific cloud masking scheme is presented in detail. The derived scene cloud covers and pixel–level statistics for 124 ASTER cases are compared to those calculated from single–band thresholds developed on a scene–by–scene basis. Subsequently the research–level ASTER cloud property retrieval algorithm is documented.

### 3.1 Cloud Detection for ASTER

Cloud detection from moderate to high resolution imagers can take on many forms, from simple single thresholding approaches to more elaborate machine learning approaches. As clearly demonstrated in Yang and Di Girolamo (2008), cloud detection algorithms must be designed with a particular purpose in mind. The retrieval algorithm presented in Section 3.2 is a research–level algorithm and is specifically employed to study the effects of sensor resolution on remote sensing products of MBL clouds. For this reason, the highest resolution available from ASTER (15 m) is targeted, while the need for an operationally complete and globally validated cloud detection algorithm is not required at this time. Still, the manually tedious effort to produce scene–by–scene cloud masks for a multitude of different resolutions based on a single (or more) threshold approach (e.g., Wielicki and Welch, 1986, Zhao and Di Girolamo, 2006) is replaced in favor of a hybrid approach. Here, individual cases are selected based on the presence of low–level water clouds over the ocean and the absence of high–level cirrus that impacts the cloud property retrieval (Wind et al., 2010). Subsequently, a simple decision-tree approach (e.g., Saunders and Kriebel, 1988; Ackerman et al., 1998) was developed, using five thresholding tests to produce a 15 m resolution cloud mask. Since the focus of this study is on the feasibility of cloud microphysical retrievals from ASTER, the cloud masking scheme is cloud–conservative. A cloud mask designation similar to MODIS is employed, namely confidently cloudy, probably cloudy, probably clear, and confidently clear. The five cloudiness tests performed are described below:

    (i) ASTER band 3N reflectances $\gamma_{0.86,A}$ need to exceed distinct thresholds. Similar tests to identify clear–sky pixels have been reported by Ackerman et al. (1998); Ackerman et al. (2008), Frey et al. (2008) and Banks and Mélin (2015) for MODIS observations, generally establishing thresholds of $\gamma_{0.86,M} < 0.03$ for confidently clear and $\gamma_{0.86,M} > 0.065$ for cloudy pixels.

    (ii) Similar to test (i), a threshold for ASTER band 5 reflectances $\gamma_{2.1,A}$ is defined to distinguish between clouds and the darker ocean surface.

    (iii) A ratio of ASTER band 3N and band 2 reflectances, calculated as $r_1 = \frac{\gamma_{0.86,A}}{\gamma_{0.65,A}}$, is applied to distinguish clouds from the darker ocean surface, as well as from measurement over land. This ratio utilizes the rather constant spectral behavior of clouds in the VNIR, which leads to their white appearance. Ackerman et al. (1998) found thresholds of $0.8 < r_1 < 1.1$, while Ackerman et al. (2008) and Banks and Mélin (2015) reported adjusted lower thresholds of $0.85$ and $0.95$ for confidently





190   clear and cloudy pixels, respectively. The upper threshold is usually set to 1.1, in part to exclude land
surfaces. Tests with different ASTER cases have shown that this value can reach values of $r_1 > 1.3$
for cloud observations, while land surfaces show $r_1 >> 1.3$. Tests (i)–(iii) are usually sufficient for
identifying reasonably bright cumulus clouds (i.e., $\gamma_{0.86,A} \geq 0.2$).

(iv) To better distinguish cloud edges and very thin cumuli from the ocean surface it proves helpful

195   to define a second ratio in the VNIR. The ratio of ASTER band 1 and band 2 reflectances, calculated
as $r_2 = \frac{\gamma_{0.52,A}}{\gamma_{0.65,A}}$, shows rather large values of $r_2 > 1.6$ over the ocean due to increased Rayleigh
scattering (i.e., the VNIR spectrum in this range has a steeper slope). Similar to $r_1$ this ratio is close
to 1 for cloudy pixels, because of their spectrally invariant behavior in the VNIR.

Categorizing pixels into confidently cloudy, probably cloudy, probably clear, and confidently clear

200   pixels is performed with the decision–tree illustrated in Figure 3. The derived thresholds for tests (i)–
(iv) are as follows: Confidently cloudy pixels (cloudiness flag '0') indicate pixels with sufficiently
large ASTER band 3N reflectances and either contain bright low level cumuli or clouds with a large
vertical extent. These pixels are identified by $\gamma_{0.86,A} > 0.065$, $\gamma_{2.1,A} > 0.02$, $0.80 < r_1 < 1.75$, and
$r_2 < 1.2$. Probably cloudy pixels (cloudiness flag '1') are associated with observations covering

205   rather thin clouds and cloud edges. They are characterized by lower band 3N reflectances. These pix-
els are identified by $\gamma_{0.86,A} > 0.03$, $\gamma_{2.1,A} > 0.015$, $0.75 < r_1 < 1.75$, and $r_2 < 1.35$. Probably clear
pixels (cloudiness flag '2') are characterized by $\gamma_{0.86,A} > 0.03$, $\gamma_{2.1,A} > 0.01$, $0.70 < r_1 < 1.75$,
and $r_2 < 1.45$. Usually, these pixels are clear. However, if such pixels are flagged as cloudy, a cloud
property retrieval either fails or yields an ASTER cloud optical thickness $\tau_A < 5$. All other pixels are

210   identified as clear (cloudiness flag '3'). These thresholds, which comprise the first step in the new
cloud masking scheme, were set through inspection of 210 ASTER MBL scenes sampled off the
Coast of California and the tropical western Atlantic (Zhao and Di Girolamo, 2006) between April
2003 and July 2007. These observations have been performed at full ASTER resolution, keeping in
mind that we are siding on a cloud–conservative cloud mask. While the thresholds are derived for a

wide range of solar zenith angles ($\theta_0 = 33.4° - 63.2°$), aerosol optical depths ($0.04 - 1.49$) and even
a small number of sun–glint cases they are static with no dependence on $\theta_0$. As demonstrated below,
the quality of the cloud masks meet the purpose of this study and are believed to be more broadly
appropriate for deep ocean scenes, in atmospheres with low aerosol turbidity, and outside of strong
sun–glint and large $\theta_0$. However, it should be noted that further refinement of these thresholds are

likely for investigations outside the scope of this study.

Due to increased horizontal photon transport in more complex broken cumulus scenes (where
there is a large number of cumuli with small horizontal extent), as well as cases with pronounced
sun glint, it is found that tests (i)–(iv) can become noisy and sometimes falsely identify clear pixels
as cloudy. Therefore, in a second step a threshold for the brightness temperature $T_{B,11}$, derived from

the ASTER Band 14 radiances, is defined to correctly label such pixels as clear–sky observations.
This threshold ($T_{B,c5}$, cloudiness test (v)) is calculated as the $5^{\text{th}}$ percentile of $T_{B,11}$ sampled over





all clear pixels (cloudiness flag '3') if the fraction of clear pixels $n_c$ in the respective scene is at least
$0.03$. This guarantees a sufficient number of samples to calculate frequency distributions of $T_B$ (e.g.,
even for a horizontal resolution of $1000\,m$ over $100$ clear pixels remain). In order to match the spatial
resolution of the VNIR observations, $T_{B,11}$ are scaled up to the VNIR resolution (i.e., each $T_{B,11}$
sample at $90\,m$ resolution is replicated onto $36$ subpixels with a horizontal resolution of $15\,m$).

Figures 2(a)–(e) show the results of the five thresholding tests for a broken cumulus case observed
over the tropical western Atlantic on 01/22/2005 at 14:48 UTC. Figures 2(a)–(b) show observa-
tions of $\gamma_{0.86,A}$ and $\gamma_{2.1,A}$ over a multitude of small cumuli and the ocean surface. The surface
samples exhibit $\gamma_{0.86,A} \leq 0.03$ and $\gamma_{2.1,A} \leq 0.008$, whereas the thick parts of the cumuli are charac-
terized by $\gamma_{0.86,A} > 0.1$ and $\gamma_{2.1,A} > 0.015$. Meanwhile, over cloud edges and very thin cloud parts
$\gamma_{0.86,A} < 0.1$ and $\gamma_{2.1,A} < 0.015$ are observed. Figures 2(c)–(d) illustrate $r_1$ and $r_2$, respectively.
As mentioned earlier, these ratios show values around $1$ for cloudy pixels, while the ocean can be
clearly discriminated with values of $r_1 \leq 0.7$ and $r_2 \geq 1.45$. Results for $T_{B,11}$, shown in Figure 2(e),
illustrate a decrease in derived brightness temperatures for cloudy pixels compared to the ocean sur-
face in the range of $(2-3)\,K$. Finally, Figure 2(f) shows the derived cloud mask for the example
case sampled on 01/22/2005, yielding reliable results compared to the observations of $\gamma_{0.86,A}$.

A comparison between calculated scene cloud covers $C_A$ based on the cloud masking scheme
reported in Zhao and Di Girolamo (2006), which utilizes a single case–by–case threshold for $\gamma_{0.86,A}$,
and those based on cloudiness tests (i)–(v) show a high agreement. A frequency distribution of the
difference in scene cloud covers between the case–by–case threshold and the new cloud masking
scheme ($\Delta C_A$) is shown in Figure 4. Derived $\Delta C_A$ are in the range of $\Delta C_A = -0.07-0.10$, with a
median difference amounting to an underestimation of about $0.004$ and an interquartile range (IQR)
of $0.019$. These maximum deviations, however, are only observed for a small number of cases. These
are characterized by either strong sun glint, which makes it difficult to reliably detect all clouds
with just a single threshold for $\gamma_{0.86,A}$, or by a complex cloud structure with pronounced horizontal
photon transport, which yields some false cloudy pixel designations by the single–threshold scheme.
For these cases cloudiness test (v) assures that the new ASTER cloud mask algorithm produces more
reliable results. The majority of scenes ($90.4\%$) are characterized by a good agreement in estimated
cloud amount in the range of $-0.04 \geq \Delta C_A \leq 0.04$. The slight skew towards positive $\Delta C_A$ values
is consistent with the cloud conservative goal of the new automated algorithm for the purpose of this
study. On the pixel level it is found that of all cloudy pixels, as determined by the $\gamma_{0.86,A}$ threshold
introduced in Zhao and Di Girolamo (2006), $80.8\%$ are also identified by the new cloud masking
scheme, about $14.6\%$ are missed but have no successful cloud property retrieval, $0.03\%$ are missed
and have a retrieved cloud optical thickness $\tau_A \geq 5$, and $4.6\%$ are missed and exhibit $\tau_A < 5$. Of all
clear pixels, as determined by the single–band threshold, $99.4\%$ are also identified as clear by the
new cloud masking scheme, $0.2\%$ are characterized as cloudy with a failed cloud property retrieval,
and $0.4\%$ exhibit a cloudy designation and $\tau_A < 1$.



### 3.2 Retrieval Algorithm

After cloud masking a retrieval of cloud top, optical and microphysical properties is performed. The research–level ASTER retrieval setup uses the same algorithms as the operational MODIS C6 retrieval, which provides the means to execute the same retrieval code for ASTER and eliminates uncertainties when comparing retrieval products between the different sensors. This allows for a comprehensive comparison between the MODIS and ASTER results without biases due to the ap-

plied set of equations. It also allows for the use of well tested and documented code.

#### 3.2.1 Cloud Top Properties

The retrievals of ASTER cloud top pressure, cloud top temperature and cloud top height are performed using the optimal estimation method in conjunction with the operational MODIS C6 IR window retrieval. This precise algorithm combination is used with great success for the operational

retrievals of cloud top properties for the MSG–SEVIRI imager (Hamann et al., 2014). Data input is provided by the collected radiances $I_A$ in combination with the profiles of atmospheric temperature, moisture, ozone, and surface temperature. The current implementation of the ASTER retrieval uses Global Data Assimilation System (GDAS) 1–degree analysis from the National Centers for Environmental Prediction (NCEP) for this purpose (Derber et al., 1991). The surface emissivity data comes

from the broadband spectral emissivity database produced for the MOD07 atmospheric profiles product (Seemann et al., 2008). To account for the presence of possible snow or sea ice in the scene the NCEP sea ice product (Hunke and Dukowicz, 1997) is used together with the National Snow and Ice Data Center (NSIDC) $27\,\mathrm{km}$ resolution 5–day running average land snow cover (Nolin et al., 1998). The retrieval begins by obtaining the profiles of IR transmittance and radiance for the given ancillary

atmospheric and surface parameters at the specific pixel. The calculations are performed using the Pressure–layer Fast Algorithm for Atmospheric Transmittance (PFAAST) code (Strow et al., 2003). PFAAST is also implemented in the operational MODIS cloud top properties retrieval algorithm documented in Baum et al. (2012), except for the ASTER retrievals the full ASTER SRFs are used instead of MODIS ones. The cloud thermodynamic phase is subsequently computed using the bis-

pectral IR method based on the brightness temperature difference between the $8.5\,\mu\mathrm{m}$ and $11\,\mu\mathrm{m}$ bands. The method is identical to the one used by the operational MODIS C5.1 IR cloud thermodynamic phase retrieval (Baum et al., 2000). After determining the thermodynamic phase, the retrievals of cloud top pressure, cloud top temperature and cloud top altitude are performed assuming unity cloud emissivity as an initial guess. Actual values are derived from the optimal estimation algo-

rithm, which is also used by the MODIS–VIIRS data continuity product for cloud top properties (Heidinger et al., 2014). If the calculated cloud top pressure is larger than $650\,\mathrm{mb}$ the operational MODIS C6 IR window retrieval algorithm is used to calculate the final value of cloud top pressure (Baum et al., 2012). Cloud phase is also corrected as necessary based on cloud top temperature and





cloud top pressure provided by the optimal estimation algorithm. If prior to the optimal estimation
calculations the cloud phase was identified as liquid water, but the cloud top temperature is less than
$245\,\mathrm{K}$ or cloud top pressure is less than $375\,\mathrm{mb}$, the cloud phase value is changed to ice.

### 3.2.2 Cloud Optical and Microphysical Properties

The retrievals of cloud optical thickness $\tau_A$ and effective droplet radius $r_{\mathrm{eff},A}$ are based on the bis-
pectral retrieval approach, which applies atmospherically corrected cloud top reflectances at two
distinct wavelength bands and utilizes retrieval lookup tables (LUT) (Twomey and Seton, 1980;
Nakajima and King, 1990; Rossow and Schiffer, 1991). This approach uses the distinct sensitivities
of reflectances in the VNIR to $\tau$ and reflectances in the SWIR to $r_{\mathrm{eff}}$ (Marshak et al., 2006). ASTER
bands 3N and 5 provide the VNIR and SWIR reflectances, respectively. Similar to the retrieval of
cloud top properties, the ASTER retrieval uses the same algorithms as the operational MODIS C6 re-
trievals described in King et al. (1997), Platnick et al. (2003), and MODIS Characterization Support Team
(2012).

Atmospheric correction is performed by generating two–way atmospheric transmittance tables
containing the effects of water vapor and molecular absorption by various gases (Platnick et al.,
2003; Wind et al., 2010). Simulations are done with the moderate resolution atmospheric transmis-
sion (MODTRAN) code version 4.2r1 (Berk et al., 1998) for the complete ASTER VNIR and SWIR
range (considering the full SRF of each band). The standard atmosphere in the MODTRAN in-
put is modulated by the averaged clear–sky profiles from the European Centre for Medium–Range
Weather Forecasts (ECMWF) Re–analysis (ERA–40) database (Chevallier, 2002). Band 2 and 3N
reflectances require a correction for above–cloud ozone amount following the method described
in Platnick et al. (2003) for the operational MODIS C6 retrieval algorithm. Here, the below–cloud
ozone amount is assumed to be negligible and the total column ozone variable (TOZNE) of the
NCEP GDAS is used as input. Once all corrections are applied, the surface contribution is removed
from the measured ASTER reflectance. For that purpose the gap–filled MODIS surface albedo prod-
uct is used (Moody et al., 2005, 2007, 2008) for retrievals over land. When retrievals are performed
over ocean, the NCEP GDAS variables U10M and V10M are used to derive the value of wind speed.
This wind speed is used as input in the Cox–Munk model to obtain the ocean surface reflectance
(Cox and Munk, 1954a, b). Similar to the corrections in the cloud top retrievals, the NSIDC land
snow cover and NCEP sea ice product are used to account for the presence of snow or sea ice
in the land albedo and ocean surface reflectance. The estimated snow and ice fractions, together
with the statistical ecosystem–based MODIS spectral snow and ice albedo product (Moody et al.,
2007) and ecosystem type from the International Geosphere–Biosphere Programme (IGBP) dataset
(Loveland et al., 2000), provide the means to estimate the final value of surface albedo.

The interpolation of the VNIR and SWIR reflectances is performed in different LUTs to accom-
modate the differences in the band centers and SRFs between ASTER and MODIS (see Figure





1). LUTs were generated with the discrete ordinates radiative transfer (DISORT) model developed
by Stamnes et al. (1988, 2000), and the computations were carried out with $64$ streams to capture
both upwelling and downwelling radiance (32 up and 32 down). The wind speed–dependent bidi-
rectional surface reflectance of the ocean is parameterized following Cox and Munk (1954a, b), as
implemented in the radiative transfer library libRadtran (Mayer and Kylling, 2005; Mayer, 2009).

The single scattering properties of liquid water clouds were computed from Mie Theory according
to Wiscombe (1980), assuming a Modified Gamma droplet size distribution with an effective vari-
ance of $0.10$. The LUTs do not include the additional contributions from Rayleigh scattering, which
are added to the atmospherically corrected ASTER reflectances before a retrieval is attempted. The
added amount of Rayleigh scattering is a function of cloud top pressure and is accounted for dynam-

ically, using the retrieved value of cloud top pressure as described in Wang and King (1997). For
both MODIS and ASTER, the retrieved $\tau$ is scaled to the respective $0.65\,\mu m$ band (i.e., band 1 for
MODIS and band 2 for ASTER).

It must be noted that for the cloud property retrieval at $15\,m$ horizontal resolution SWIR re-
flectances are scaled up to match the resolution of band 3N (i.e., each SWIR reflectance sample

at $30\,m$ resolution is replicated onto 4 subpixels with a horizontal resolution of $15\,m$). This intro-
duces uncertainties in the retrieved cloud parameters at the highest ASTER resolution. As described
in Section 5.4 these uncertainties are estimated to be $\pm 0.5$ (for $\tau_A$) and $\pm 0.7\,\mu m$ (for $r_{eff,A}$).

King et al. (1997) and Platnick et al. (2004) discussed the retrieval uncertainties associated with
MODIS cloud products, which are the result of instrument errors, uncertainties in the radiometric

calibrations and the applied radiative transfer model, as well as ancillary data sets used as input for
the atmospheric correction algorithm, among other components. The current MODIS retrieval prod-
ucts provide pixel–level uncertainty estimates for $\tau$ and $r_{eff}$. Because the ASTER retrieval algorithm
deploys the same retrieval code, ASTER pixel–level retrieval uncertainties are derived in a similar
way. An approximate, albeit less comprehensive, uncertainty range due to radiometric uncertain-

ties, only, can be estimated by applying the individual measurement uncertainties $\delta$ of the simulated
reflectances $\gamma_{LUT}$ in the VNIR and SWIR by:

$$\begin{aligned}
\Delta\tau_{LUT}(\gamma_{LUT}) &= \frac{\tau_{LUT}(\gamma_{LUT}) - \tau_{LUT}(\gamma_{LUT} \pm \delta)}{\tau_{LUT}(\gamma_{LUT})}, \\
\Delta r_{eff,LUT}(\gamma_{LUT}) &= \frac{r_{eff,LUT}(\gamma_{LUT}) - r_{eff,LUT}(\gamma_{LUT} \pm \delta)}{r_{eff,LUT}(\gamma_{LUT})}.
\end{aligned} \tag{3}$$

Here, $\delta$ can either increase or decrease the actually observed $\gamma_{LUT}$. Calculating $\Delta\tau_{LUT}$ and $\Delta r_{eff,LUT}$

for each possible combination of $\gamma_{LUT} \pm \delta$ in the VNIR and SWIR yields an expected uncertainty
range for the retrieved cloud properties. Assuming $\delta < 4\,\%$ (Yamaguchi et al., 1998) and including
results of $r_{eff,LUT} > 4\,\mu m$ and $\tau_{LUT} > 4$, only, yields mean retrieval uncertainties of $\Delta\tau_{LUT} = 0.15$
and $\Delta r_{eff,LUT} = 0.23$, respectively.





### 3.3 LUT Differences due to SRF Differences

ASTER–specific LUTs have been developed, which are used to retrieve the cloud optical thickness $\tau_A$ and $r_{\text{eff,A}}$. Figure 5(a) illustrates simulated reflectances in the VNIR as a function of input cloud optical thickness $\tau_{\text{LUT}}$ for ASTER ($\gamma_{0.86,\text{LUT,A}}$, black) and MODIS ($\gamma_{0.86,\text{LUT,M}}$, green). The simulations resemble 1D reflectance calculations over a model cloud over the ocean. The input solar and viewing geometry is based on example scene C19, discussed in Table 2 and Section 4.2, which

yields solar zenith and azimuth angles of $\theta_0 = 24°$ and $\varphi_0 = 143°$, while the sensor zenith angles are $\theta_s = 8.59°$ (VNIR) and $\theta_s = 8.54°$ (SWIR). Simulations have been performed for two constant values of input cloud effective droplet radius $r_{\text{eff,LUT}}$ (highlighted by different symbols). The relationship between VNIR reflectance and $\tau_{\text{LUT}}$ exhibits the well–known monotonically increasing, concave behavior for both sensors and there is little difference between ASTER and MODIS, as well

as between the different $r_{\text{eff,LUT}}$. There is a decrease in VNIR $\gamma_{\text{LUT}}$ with increasing $r_{\text{eff,LUT}}$, which was also stated in Marshak et al. (2006). Figure 5(c) shows the theoretical scale factor $f_{0.86,\text{LUT}}$, defined as:

$$f_{0.86,\text{LUT}}(\tau_{\text{LUT}}, r_{\text{eff,LUT}}) = \frac{\gamma_{0.86,\text{LUT,A}}(\tau_{\text{LUT}}, r_{\text{eff,LUT}})}{\gamma_{0.86,\text{LUT,M}}(\tau_{\text{LUT}}, r_{\text{eff,LUT}})}. \tag{4}$$

This means that $f_{0.86,\text{LUT}}$ is the ratio of ASTER to MODIS VNIR reflectance for each $(\tau_{\text{LUT}}, r_{\text{eff,LUT}})$

pair. The behavior of $f_{0.86,\text{LUT}}$ for the given solar and viewing geometry and constant values of $r_{\text{eff,LUT}}$ illustrates that for the same $\tau_{\text{LUT}}$ ASTER reflectances are about $1 - 2\%$ brighter than the MODIS reflectances. Averaging $f_{0.86,\text{LUT}}$ over all $\tau_{\text{LUT}}$ and $r_{\text{eff,LUT}}$ combinations yields a mean value of 1.005, illustrating that the VNIR bands of both sensors are directly comparable. For very thin clouds ($\tau_{\text{LUT}} < 2$) calculated scale factors reach values as high as 1.099, while there is only a

small dependence on $r_{\text{eff,LUT}}$ with values in the range of $0.981 - 1.019$.

Figure 5(b) illustrates the monotonically decreasing, convex behavior of simulated SWIR reflectances $\gamma_{2.1,\text{LUT,A}}$ (ASTER) and $\gamma_{2.1,\text{LUT,M}}$ (MODIS) as a function of $r_{\text{eff,LUT}}$. Due to differences in the respective SRF simulated $\gamma_{2.1,\text{LUT,A}}$ in the SWIR band are higher than the corresponding MODIS $\gamma_{2.1,\text{LUT,M}}$, with increased reflectances for optically thicker clouds. The theoretical

scale factors $f_{2.1,\text{LUT}}$, derived from the SWIR reflectances similar to Eq.(4) and shown in Figure 5(d), cover a range of $0.99 - 1.23$ with a mean value of 1.092. This implies that for the same $\tau_{\text{LUT}}$ and $r_{\text{eff,LUT}}$ the respective ASTER observation in the SWIR band is brighter than the MODIS measurement.

## 4 Examples of High–resolution Retrievals

This section introduces all ASTER MBL scenes used in this study. Moreover, examples of the retrieved cloud optical thickness and effective droplet radius based on high–resolution ASTER reflectance measurements are presented.





### 4.1 Data Set

Since the goal of this study is to examine the feasibility of ASTER cloud property retrievals in
comparison to MODIS, a sufficient number of samples of both fully and partially cloudy pixels at
1000 m scales is required. For this reason, the 124 ASTER scenes collected over the tropical western
Atlantic (used in evaluating the 15 m cloud mask) are not sufficient. That dataset was populated
entirely of trade wind cumuli with a peak in the cloud fraction distribution at $(400 - 500)$ m in cloud
equivalent diameters (Zhao and Di Girolamo, 2007). As a result, a new data set is introduced, which
consists of scenes with much more extensive MBL cloud cover and cloud sizes. The data set in this
study consists of 48 MBL cloud cases sampled over the Pacific Ocean off the Coast of California,
sampled between 05/2003 and 07/2007. Granules were manually chosen to include MBL clouds and
that resemble altocumulus or broken cumulus scenes. The number of available cases is constraint by
the availability of co–located ASTER and MODIS data with successful cloud property retrievals at a
horizontal resolution of 1000 m (which excludes some broken cumulus scenes). Moreover, selected
scenes are characterized by the absence of overlying cirrus, complex multi–layered cloud systems,
and pixels with ice phase. It was made sure that the cases sampled in 2007 are not affected by
the reduced dynamic range of the ASTER SWIR band signal, which started to affect the ASTER
data starting mid 2007. The area covered by the 48 MBL scenes is embedded within $125.924° \mathrm{W} -$
$117.038° \mathrm{W}$ and $32.051° \mathrm{N} - 44.427° \mathrm{N}$.

Table 2 lists the case numbers (C1–C48), as well as the sample date of each scene. A wide range
of different scene characteristics are covered, with estimated domain–averaged cloud covers $C_M$,
based on MODIS cloud flags '0' and '1' (i.e., 'confidently' and 'probably cloudy' pixels), between
$C_M = 0.01 - 1.00$. There are 25 scenes with $C_M = 0.75 - 0.99$, 9 scenes with $C_M = 0.25 - 0.74$,
and 9 scenes with $C_M < 0.25$. Completely overcast conditions (i.e., $C_M = 1.00$) are found for 18
scenes. The solar zenith angle for these cases varies between $\theta_0 = 17.96° - 63.84°$, with 24, 34, and
4 scenes having $\theta_0 < 30°$, $30° \geq \theta_0 < 60°$, and $\theta_0 \geq 60°$, respectively.

### 4.2 High–resolution Retrievals

Two ASTER scenes are selected as case studies to demonstrate the feasibility of high–resolution
ASTER retrievals and to highlight the differences between the co–located ASTER and MODIS
retrieval products.

The first scene (C14) is an altocumulus field with a domain–averaged cloud cover of $C_M = 1.0$
sampled at $19:15$ UTC on 05/13/2003. The solar geometry is characterized by $\theta_0 = 24°$ and $\varphi_0 =$
$143°$, placing the Sun's position in the South–West of the scene. Figure 6(a) shows the single band
grayscale image of $\gamma_{0.86,A}$. $\gamma_{0.86,A}$ have been sampled with a horizontal resolution of 15 m, which
allows for the detection of small–scale cloud inhomogeneities and dynamically induced, cell–like
cloud structures. In contrast, the single band grayscale image of MODIS VNIR reflectances $\gamma_{0.86,M}$,





shown in Figure 6(b), appears visibly smoother due to the horizontal resolution of the measurements of 1000 m.

A second, significantly more inhomogeneous scene (C19) with $C_M = 0.88$ is shown in Figures 6(c)–(d), illustrating single band grayscale images of $\gamma_{0.86,A}$ and $\gamma_{0.86,M}$, respectively. C19 was sampled at $19:20$ UTC on 06/10/2005. The solar and viewing geometry is similar to C14 with $\theta_0 = 20°$ and $\varphi_0 = 136°$. The cloud field is characterized by an increased heterogeneity and while the cloud cover is rather high, larger areas containing thin cloud pixels are visible throughout the

scene. Contrary to the ASTER measurements, $\gamma_{0.86,M}$ for the rather thin parts in the middle of the granule seem to be very low and the numerous delicate cloud structures (e.g., between $125.400°$ W – $125.100°$ W and $38.700°$ N – $38.900°$ N) are smoothed out.

Figures 7(a)–(b) show the cloud optical thickness retrieved from ASTER ($\tau_A$) and MODIS ($\tau_M$) reflectances sampled above scene C14 on 05/13/2003. The presented MODIS results include par-

tially cloudy pixels. Observed $\tau_A$ are about $15 - 18$ for the thick cloud parts and dip to around 8 between the cell structures. For the brightest cloud sections $\tau_A$ reaches values of 23. Similar observations can be made from the MODIS retrieval, with $\tau_M = 15 - 18$ for the thicker cloud parts and reduced $\tau_M \approx 8$ for the intermittent sections between the cell structures. As with the ASTER retrievals, there are occasional observations of $\tau_M > 20$ for the thickest cloud parts in the south of

the granule. However, the rather interesting, fuzzy behavior of $\tau_A$ around individual cells, especially visible in the North–East of the granule, is smoothed by the MODIS observations.

Retrieval results for C19 are illustrated in Figures 7(c)–(d), showing $\tau_A$ and $\tau_M$, respectively. A significant number of cloud holes are embedded within larger areas of thin cloudy pixels, where the optical thickness observations can be as low as $\tau_A = 2$. Thicker cloud parts include a number of

samples with $\tau_A \approx 15$, reaching values of about $\tau_A = 25$ at its brightest points. Observed $\tau_M$ for this scene are again comparable to the ASTER results, although there are visibly more MODIS pixels throughout the granule where the retrieval fails.

Results of the effective droplet radius retrieval from ASTER ($r_{eff,A}$) and MODIS data ($r_{eff,M}$) are illustrated in Figure 8. For the example scene C14, shown in Figures 8(a)–(b), the effective

radius retrieval shows a very homogeneous distribution, with the majority of observations around $r_{eff,A} = (7-9)\,\mu m$, which is close to a mono–disperse $r_{eff,A}$ field. Likewise, the MODIS retrieval shows that most results are between $r_{eff,M} = (6-9)\,\mu m$, revealing a good agreement between both sensors. However, there is also visible striping in the $r_{eff,M}$ results, which is caused by electronic crosstalk between various MODIS bands (Xiong et al., 2003, 2009; Sun et al., 2010, 2014).

Figures 8(c)–(d) illustrate retrieved $r_{eff,A}$ and $r_{eff,M}$ for C19. Similar to the $\tau_A$ and $\tau_M$ results there is a high degree of heterogeneity in observed effective droplet radii, which exhibit a range of $r_{eff,A} = (7-28)\,\mu m$. Some of the largest $r_{eff,A} > 20\,\mu m$ are sampled around the thinnest cloud parts, as well as cloud holes, in the North–West of the granule (around $39.00°$N and $125.2°$W). This implies a possible impact of clear–sky contamination and 3D radiative effects. A similar behavior is





observed in the $r_{\mathrm{eff,M}}$ field, although the number of failed retrievals is significantly higher compared to the ASTER data. The delicate cloud structures throughout the scene are characterized by small–scale fluctuations in $r_{\mathrm{eff,A}}$ between $r_{\mathrm{eff,A}} = (10-20)\,\mu\mathrm{m}$. The smaller horizontal resolution of the MODIS observations does not capture these finer cloud structures.

## 5 Comparison of ASTER and MODIS Results

In this section a statistical comparison between the operational MODIS C6 retrieval products with co–located ASTER results is presented, first for the two case studies introduced in Section 4.2, and subsequently on a statistical basis for all 48 MBL cloud scenes.

### 5.1 Aggregation

To yield a true comparison of reflectances and retrieved cloud variables, the high–resolution ASTER
digital counts $d_{\mathrm{A}}(\Delta\lambda)$ are aggregated within each $(1000 \cdot 1000)\,\mathrm{m}$ MODIS pixel. This requires the definition of pixel corners for each ASTER and MODIS observation. In a first step, high–resolution ASTER geolocation information is derived by interpolating the Geometric Correction Tables that are included as 11x11 arrays in the structural metadata of each ASTER data container. This yields latitude and longitude values for each individual sample and band. Subsequently, the four corners
of each ASTER pixel are defined by triangulation between the neighboring geolocation data points. Similar analysis provides the respective corners of each MODIS pixel.

Figure 9(a) illustrates the derived pixel dimensions (grey lines) for all MODIS observations from scene C19. For two example MODIS pixels, all co–located ASTER VNIR (blue lines) and SWIR (red lines) pixels are shown. A close–up of these two MODIS pixels is given in Figure 9(b).
For the aggregation of digital ASTER counts $d_{\mathrm{A}}(\Delta\lambda)$ within a MODIS pixel, an ASTER sample is included if any of its four corners lies within a respective MODIS pixel. Taking into account the different spatial resolutions of both instruments, $d_{\mathrm{A}}(\Delta\lambda)$ from over 4400 and 1100 individual ASTER VNIR and SWIR pixels are aggregated within each MODIS pixel, respectively. It is important to note that $d_{\mathrm{A}}(\Delta\lambda)$ from ASTER samples at the edge of the respective MODIS pixel, which
are only partially within a MODIS pixel's boundaries, are not weighted according to the covered area. This yields uncertainties in the aggregated digital counts of $< 0.05$. Also, the grayscale images in Figure 6(a) and 6(c) reveal that there are a number of rows of samples with $d_{\mathrm{A}}(\Delta\lambda) = 0$ at the left and right edges (i.e., West and East) of the ASTER domain. The same is true for the upper and lower edges (i.e., North and South), although significantly less samples are affected. MODIS pixels
including any of these edge pixels (at the native ASTER resolution) are omitted from the analysis.

The aggregated digital counts are subsequently used to derive aggregated VNIR and SWIR ASTER reflectances $\gamma_{0.86,\mathrm{AaM}}$ and $\gamma_{2.1,\mathrm{AaM}}$, which provide the input for the cloud property retrieval. This yields ASTER cloud optical thicknesses $\tau_{\mathrm{AaM}}$, effective droplet radii $r_{\mathrm{eff,AaM}}$, and atmospherically





corrected VNIR and SWIR reflectances $\widehat{\gamma}_{0.86,\mathrm{AaM}}$ and $\widehat{\gamma}_{2.1,\mathrm{AaM}}$. Here, the subscript 'AaM' refers to

'ASTER aggregated in MODIS'.

### 5.2  Case Studies

#### 5.2.1  Reflectance Comparison for C14 and C19

Figure 10(a) shows a comparison between atmospherically corrected VNIR reflectances sampled by MODIS ($\widehat{\gamma}_{0.86,\mathrm{M}}$) and co–located $\widehat{\gamma}_{0.86,\mathrm{AaM}}$ for the homogeneous scene C14. The data set is

sorted into overcast and partially cloudy pixels, respectively, determined from the subpixel cloud cover $C_{\mathrm{sub}}$ based on ASTER cloudiness flags '0' and '1' at $15\,\mathrm{m}$ horizontal resolution. Overcast pixels exhibit $C_{\mathrm{sub}} = 1.0$, while partially cloudy pixels are characterized by $C_{\mathrm{sub}} < 1.0$. Only pixels containing liquid water clouds and both a successful MODIS and ASTER cloud property retrieval are included in the analysis. For the sake of display, only, the scale factor $f_{0.86,\mathrm{LUT}}$ between the

ASTER and MODIS LUT reflectances in the VNIR is calculated for the respective C14 geometry and ($\tau_{\mathrm{AaM}}$, $r_{\mathrm{eff,AaM}}$) pair and multiplied with $\widehat{\gamma}_{0.86,\mathrm{AaM}}$. This accounts for the theoretical differences in the respective SRF between both instruments (see Section 3.3). A strong positive correlation with $R = 0.993$ is observed and all observations lie close to the 1:1 line. However, for C14 sampled $\widehat{\gamma}_{0.86,\mathrm{AaM}}$ appear to be slightly brighter than $\widehat{\gamma}_{0.86,\mathrm{M}}$, which would result in larger $\tau_{\mathrm{AaM}}$.

The correlation between $\widehat{\gamma}_{2.1,\mathrm{M}}$ and $\widehat{\gamma}_{2.1,\mathrm{AaM}}$ sampled in the SWIR (and multiplied with the respective $f_{2.1,\mathrm{LUT}}$) is shown in Figure 10(b). Similar to the VNIR observations, there is a high agreement with $R = 0.938$.

Similar scatter plots for atmospherically corrected reflectances in the VNIR and SWIR sampled over the more inhomogeneous scene C19 are shown in Figures 10(c)–(d). Even higher correlation

coefficients of $R = 0.996$ and $R = 0.988$ are observed, respectively, and overall there is a good agreement between the MODIS and co–located ASTER observations. While reflectances from both instruments are mostly comparable, $\widehat{\gamma}_{0.86,\mathrm{M}}$ ($\widehat{\gamma}_{2.1,\mathrm{M}}$) are slightly larger than $\widehat{\gamma}_{0.86,\mathrm{AaM}}$ ($\widehat{\gamma}_{2.1,\mathrm{AaM}}$) for brighter pixels.

For all overcast pixels sampled in C14 and C19, the remaining bias between $\widehat{\gamma}_{0.86,\mathrm{AaM}}$ and

$\widehat{\gamma}_{0.86,\mathrm{M}}$, after correcting for the theoretical difference due to their respective SRF, is about $3.5\%$. Likewise, the remaining bias between $\widehat{\gamma}_{2.1,\mathrm{AaM}}$ and $\widehat{\gamma}_{2.1,\mathrm{M}}$ is about $0.3\%$. These values are in the range of the intercomparison results by Uprety et al. (2013), who reported radiometric bias uncertainties between the Suomi National Polar-Orbiting Partnership (Suomi–NPP) Visible Infrared Imaging Radiometer Suite (VIIRS) and MODIS in the range of $2-3\%$ for the VNIR and SWIR bands.

It is important to note that small differences in cloud top reflectances could result in possibly large differences in retrieved cloud properties.

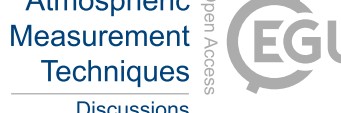

### 5.2.2 Retrieved Cloud Property Comparison for C14 and C19

There is a good agreement between the ASTER (at native resolution) and MODIS retrieval results shown in Figures 7 – 8, with both instruments covering a similar value range and spatial distribution for $\tau_{\mathrm{A}}$ and $\tau_{\mathrm{M}}$, as well as $r_{\mathrm{eff,A}}$ and $r_{\mathrm{eff,M}}$. To gain a better understanding about the difference in retrieved cloud properties from aggregated ASTER reflectances and the respective MODIS C6 products, scatterplots of the difference $\tau_{\mathrm{AaM}} - \tau_{\mathrm{M}}$ versus the difference $r_{\mathrm{eff,AaM}} - r_{\mathrm{eff,M}}$ are shown in Figure 11(a) for the homogeneous example cases C14. Samples over overcast pixels are illustrated by black circles, while gray circles indicate observations over partially cloudy pixels. Again, only pixels containing liquid water clouds and both a successful MODIS and ASTER retrieval are considered in the analysis.

Overall, there is a high agreement between the retrieved cloud properties from ASTER and MODIS, with minimum and maximum differences between $\tau_{\mathrm{AaM}}$ and $\tau_{\mathrm{M}}$ of $-0.85$ and $1.82$. Similarly, the observed minimum and maximum differences between $r_{\mathrm{eff,AaM}}$ and $r_{\mathrm{eff,M}}$ are $-0.70\,\mu\mathrm{m}$ and $1.09\,\mu\mathrm{m}$. The median difference in retrieved optical thickness (effective droplet radius) is $0.87$ ($0.13\,\mu\mathrm{m}$) with an IQR of $0.43$ ($0.48\,\mu\mathrm{m}$). For C14 there seems to be a slight bias in retrieved $\tau_{\mathrm{A}}$ of about $0.7$, resulting from the higher $\widehat{\gamma}_{0.86,\mathrm{AaM}}$ observed in Figure 10(a).

To relate these differences to the ASTER retrieval uncertainties derived in Section 2.3, the differences $\tau_{\mathrm{AaM}} - \tau_{\mathrm{M}}$ and $r_{\mathrm{eff,AaM}} - r_{\mathrm{eff,M}}$ are normalized by $\tau_{\mathrm{AaM}}$ and $r_{\mathrm{eff,AaM}}$, respectively. The results for C14 are shown in Figure 11(b), where the gray box indicates the retrieval uncertainty for both cloud variables due to radiometric uncertainties, only (see the discussion in Section 3.2.2). It is obvious that the differences in retrieved optical thickness and effective droplet radius between ASTER and MODIS are well within the retrieval uncertainties of ASTER. The best agreement between the two sensors is achieved for bright cloudy pixels where $\tau_{\mathrm{M}}, \tau_{\mathrm{AaM}} \geq 14$. Here, differences in retrieved optical thickness are in the range of $5\%$, while differences in retrieved effective droplet radius are $\pm 10\%$. With lower $\gamma_{\mathrm{AaM}}$ the retrieval differences, as well as the bias in $\tau_{\mathrm{AaM}}$, increase.

The comparison of retrieved cloud properties for the more inhomogeneous example scene C19 is shown in Figures 11(c)–(d). Differences between $\tau_{\mathrm{AaM}}$ and $\tau_{\mathrm{M}}$ range from $-3.11$ to $+1.26$ (with a median difference of $0.30$ and an IQR of $0.55$), while overcast pixels exhibit differences between $r_{\mathrm{eff,AaM}}$ and $r_{\mathrm{eff,M}}$ in the range of $-6.16\,\mu\mathrm{m}$ to $4.57\,\mu\mathrm{m}$ (with a median difference of $0.53\,\mu\mathrm{m}$ and an IQR of $1.64\,\mu\mathrm{m}$). The largest differences in retrieved effective droplet radius between ASTER and MODIS are observed for partially cloudy pixels, where $\tau_{\mathrm{M}}$ and $\tau_{\mathrm{AaM}}$ are low. Here, the difference $r_{\mathrm{eff,AaM}} - r_{\mathrm{eff,M}}$ can be as large as $-12.36\,\mu\mathrm{m}$. All these pixels are characterized by low $\tau_{\mathrm{M}}, \tau_{\mathrm{AaM}}$ and the subpixel cloud cover $C_{\mathrm{sub}}$, derived from the original $15\,\mathrm{m}$ ASTER resolution for cloudiness flags '0' and '1' (i.e., 'confidently' and 'probably cloudy' pixels), can be as low as $0.730$. This implies that the retrieval is contaminated by low ocean surface reflectance observations. While there seemed to be a positive $\widehat{\gamma}_{0.86,\mathrm{M}}$ and $\widehat{\gamma}_{2.1,\mathrm{M}}$ bias for large reflectances, the optical thickness and



effective droplet radius differences show no such bias (i.e., they are centered around $\tau_{\mathrm{AaM}} - \tau_{\mathrm{M}} = 0$ and $r_{\mathrm{eff,AaM}} - r_{\mathrm{eff,M}} = 0\,\mu\mathrm{m}$, respectively).

Normalizing the retrieval differences with $\tau_{\mathrm{AaM}}$ and $r_{\mathrm{eff,AaM}}$ illustrates that again almost all observations are within the retrieval uncertainties of the ASTER instrument. However, partially cloudy pixels yield differences between $\tau_{\mathrm{AaM}}$ and $\tau_{\mathrm{M}}$ of up to $35\%$ and overestimation in $r_{\mathrm{eff,M}}$ (compared to $r_{\mathrm{eff,AaM}}$) of up to $130\%$. Similar to C14, the best agreements between ASTER and MODIS cloud variables are achieved for bright pixels with high $\widehat{\gamma}_{0.86,\mathrm{AaM}}$ and $\widehat{\gamma}_{0.86,\mathrm{M}}$ (and subsequently

high $\tau_{\mathrm{AaM}}$ and $\tau_{\mathrm{M}}$). Here, the differences between both sensors are about $\pm 10\%$ for both the optical thickness and effective droplet radius. With decreasing $\tau_{\mathrm{AaM}}$ and $\tau_{\mathrm{M}}$ the differences increase up to the retrieval uncertainty of ASTER.

Overall, the correlation coefficients between $\tau_{\mathrm{AaM}}$ and $\tau_{\mathrm{M}}$ are $R = 0.992$ and $R = 0.995$ for C14 and C19, respectively. The correlation of $r_{\mathrm{eff,M}}$ and $r_{\mathrm{eff,AaM}}$ yields $R = 0.872$ for C14 and $R = 0.739$ for C19. Limiting the analysis to observations with overcast pixels increases the correlation

coefficient for the effective droplet radius comparison to $R = 0.889$ for C19.

### 5.3    Statistical Comparison for 48 MBL Cloud Scenes

#### 5.3.1    Cloud Mask Comparison

Co–located ASTER reflectances are used to get a cloud mask value for each pixel of the 48 MBL

cases, assigning the respective cloud mask flag according to the discussion in Section 3.1. To compare the domain–averaged cloud cover from ASTER observations ($C_{\mathrm{AaM}}$, derived from the aggregated ASTER radiances) with the operational MODIS results from the MOD35 data containers ($C_{\mathrm{M}}$), the fraction of pixels with a cloudiness flag value of '0' or '1' is calculated (i.e., the fraction of 'confidently' and 'probably cloudy' pixels).

Figure 12 shows a frequency distribution of the difference between the domain–averaged cloud covers from MODIS and co–located ASTER measurements. An agreement between $C_{\mathrm{M}}$ and $C_{\mathrm{AaM}}$ of $\pm 0.04$ is observed for 34 of the 48 analyzed MBL scenes (i.e., $73.9\%$), while $89.1\%$ of cases exhibit an agreement in scene cloud cover of $\pm 0.1$. Cases where the absolute difference between $C_{\mathrm{M}}$ and $C_{\mathrm{AaM}}$ is larger than $0.1$ are characterized by $C_{\mathrm{M}} = 0.11 - 0.87$ and include a substantial

number of pixels characterized by MODIS cloudiness flags '2' (i.e., 'probably clear' pixels). For these scenes, transitioning observations with cloudiness flags '2' to cloudiness flags '1' (i.e., assuming these pixels are 'probably cloudy' instead of 'probably clear' pixels) improves the agreement between the MODIS and ASTER domain–averaged cloud covers, which implies that the biggest discrepancies between $C_{\mathrm{M}}$ and $C_{\mathrm{AaM}}$ are caused by pixels with very thin clouds. Overall the me-

dian difference between $C_{\mathrm{M}}$ and $C_{\mathrm{AaM}}$ is basically 0 with an IQR of $0.03$.

On the pixel level, $91.4\%$ of the cloudy pixels, as identified by the MODIS cloudiness flags '0' and '1', are also flagged by the ASTER cloud masking scheme. Similarly, $94.0\%$ of clear MODIS





pixels are characterized as clear ASTER pixels. Of all cloudy MODIS pixels $7.1\%$ are missed by
the ASTER algorithm and are characterized by a failed cloud property retrieval, while $1.4\%$ exhibit
$0 > \tau_{\mathrm{AaM}} < 5$.

### 5.3.2   Reflectance Comparison for 48 MBL Cloud Scenes

Figure 13(a) shows probability density functions (PDF) of $\widehat{\gamma}_{0.86,\mathrm{M}}$ (green lines) and $\widehat{\gamma}_{0.86,\mathrm{AaM}}$ (black
lines) sampled in the VNIR. Data is from all 48 MBL scenes, but only overcast pixels are consid-
ered. As described in Section 5.2.1, overcast pixels are characterized by a subpixel cloud cover
$C_{\mathrm{sub}} = 1.0$. Although these data points theoretically include the operational MODIS PCL obser-
vations, practically no PCL pixels remain with the $C_{\mathrm{sub}} = 1.0$ constraint. Only pixels containing
liquid water clouds and both a successful MODIS and ASTER cloud property retrieval are consid-
ered. This yields $n = 52,254$ observations for the 48 MBL cases. For the sake of display, only, the
scale factor $f_{0.86,\mathrm{LUT}}$ between the ASTER and MODIS LUT reflectances in the VNIR is derived
for each observation and multiplied with $\widehat{\gamma}_{0.86,\mathrm{AaM}}$. Both $\widehat{\gamma}_{0.86,\mathrm{M}}$ and $\widehat{\gamma}_{0.86,\mathrm{AaM}}$ show a similar dis-
tribution with most observations between $\widehat{\gamma}_{0.86,\mathrm{M}}, \widehat{\gamma}_{0.86,\mathrm{AaM}} = 0.107 - 0.908$. ASTER observations
are slightly higher with mean $\widehat{\gamma}_{0.86,\mathrm{AaM}} = 0.501$ compared to mean $\widehat{\gamma}_{0.86,\mathrm{M}} = 0.479$.

PDFs of $\widehat{\gamma}_{0.86,\mathrm{M}}$ and $\widehat{\gamma}_{0.86,\mathrm{AaM}}$ for partially cloudy pixels are illustrated in Figure 13(b). Similar
to the definition of overcast pixels, partially cloudy pixels are characterized by subpixel cloud covers
$C_{\mathrm{sub}} < 1.0$ and include the operational MODIS PCL observations. The number of partially cloudy
pixels is slightly smaller than the number of overcast pixels with $n = 47,538$. Both ASTER and
MODIS reflectances show similar ranges of $\widehat{\gamma}_{0.86,\mathrm{M}}, \widehat{\gamma}_{0.86,\mathrm{AaM}} = 0.033 - 0.631$. Mean $\widehat{\gamma}_{0.86,\mathrm{AaM}} =$
$0.204$ are again slightly higher than mean $\widehat{\gamma}_{0.86,\mathrm{M}} = 0.176$.

Figures 13(c)–(d) show PDFs of SWIR reflectances $\widehat{\gamma}_{2.1,\mathrm{M}}$ and $\widehat{\gamma}_{2.1,\mathrm{AaM}}$ for overcast and par-
tially cloudy pixels, respectively. As for the VNIR observations, the SWIR $\widehat{\gamma}_{2.1,\mathrm{AaM}}$ are multiplied
with the respective scale factor $f_{2.1,\mathrm{LUT}}$. Both $\widehat{\gamma}_{2.1,\mathrm{M}}$ and $\widehat{\gamma}_{2.1,\mathrm{AaM}}$ for overcast pixels are character-
ized by a bimodal distribution, with values between $\widehat{\gamma}_{2.1,\mathrm{M}}, \widehat{\gamma}_{2.1,\mathrm{AaM}} = 0.074 - 0.505$ and maximum
values around $\widehat{\gamma}_{2.1,\mathrm{M}}, \widehat{\gamma}_{2.1,\mathrm{AaM}} = 0.2$ and $\widehat{\gamma}_{2.1,\mathrm{M}}, \widehat{\gamma}_{2.1,\mathrm{AaM}} = 0.35$. Mean $\widehat{\gamma}_{2.1,\mathrm{AaM}} = 0.333$, which
compares well to the mean $\widehat{\gamma}_{2.1,\mathrm{M}} = 0.331$. Observed $\widehat{\gamma}_{2.1,\mathrm{M}}$ and $\widehat{\gamma}_{2.1,\mathrm{AaM}}$ in the SWIR for partially
cloudy pixels range between $\widehat{\gamma}_{2.1,\mathrm{M}}, \widehat{\gamma}_{2.1,\mathrm{AaM}} = 0.023 - 0.474$. Mean observations agree well with
mean $\widehat{\gamma}_{2.1,\mathrm{AaM}} = 0.162$ compared to mean $\widehat{\gamma}_{2.1,\mathrm{M}} = 0.168$.

The good agreement between the ASTER and MODIS reflectances is also illustrated in the scatter
plots in Figures 13(e)–(f), for VNIR and SWIR reflectances, respectively. Here, observations over
overcast (partially cloudy) pixels are shown with black (grey) dots, while the diagonal black line
represents the 1:1 line. Derived $\widehat{\gamma}_{0.86,\mathrm{M}}$ and $\widehat{\gamma}_{0.86,\mathrm{AaM}}$ in the VNIR, as well as $\widehat{\gamma}_{2.1,\mathrm{M}}$ and $\widehat{\gamma}_{2.1,\mathrm{AaM}}$
in the SWIR, lie close to the 1:1 line with high Pearson's product–moment correlation coefficients of
$R = 0.996$ ($R = 0.991$ for overcast pixels and $R = 0.980$ for partially cloudy pixels) and $R = 0.992$
($R = 0.981$ for overcast pixels and $R = 0.983$ for partially cloudy pixels), respectively. However, the





slight slopes observed for the example cases C14 and C19 are also apparent for the complete data
set. For the VNIR and SWIR reflectances the linear fit functions yield slope values of $\alpha_{0.86} = 1.020$
and $\alpha_{2.1} = 1.036$, respectively, with offset values of $\beta_{0.86} = -0.033$ (for VNIR reflectances) and
$\beta_{2.1} = -0.014$ (for SWIR reflectances). This confirms the results found for the two example cases,
where ASTER reflectances are slightly higher than the MODIS observations at the lower end, and
vice versa for higher reflectances.

### 5.3.3    Retrieved Cloud Property Comparison for 48 MBL Cloud Scenes

PDFs of $\tau_M$ (green lines) and $\tau_{AaM}$ (black lines) from observations over all 48 MBL scenes are
shown in Figure 14(a). Similar to the reflectance comparison earlier, only overcast pixels containing
liquid water clouds and a successful MODIS and ASTER cloud property retrieval are considered
in the analysis. The distribution of $\tau_{AaM}$ agrees well with the MODIS product and, consistent with
the slightly higher VNIR reflectances, $\tau_{AaM}$ are slightly higher than $\tau_M$ with mean $\tau_{AaM} = 12.29$,
compared to mean $\tau_M = 11.42$. Although observations can reach values as high as $\tau_M = 90.17$, over
99.9% of pixels exhibit $1.37 \geq \tau_M, \tau_{AaM} \leq 30.00$. Restricting the analysis to partially cloudy pixels
only, shown in Figure 14(b), yields a narrower distribution with retrieved $\tau_M, \tau_{AaM} = 0.17 - 24.92$.
Here, the center of each distribution is significantly reduced from the overcast data set with mean
$\tau_{AaM} = 3.62$ and mean $\tau_M = 3.08$.

The distributions of $r_{eff,M}$ and co–located $r_{eff,AaM}$ for overcast pixels are shown in Figure 14(c),
illustrating a good agreement between both instruments. While the retrievals can be as low as
$r_{eff,M}, r_{eff,AaM} = 4.76\,\mu m$, the upper limit for both the ASTER and MODIS retrieval is a fixed
value of $r_{eff,M}, r_{eff,AaM} = 30.00\,\mu m$. For larger droplets the LUTs converge and the retrieval re-
sults become unreliable. For all 48 MBL scenes the mean $r_{eff,AaM} = 10.07\,\mu m$, which compares
well with the mean observed $r_{eff,M} = 9.93\,\mu m$. Figure 14(d) shows the PDFs of $r_{eff,M}$ and $r_{eff,AaM}$
for partially cloudy pixels. Although there seem to be more observations in the range $10.00\,\mu m \geq$
$r_{eff,M}, r_{eff,AaM} \leq 20.00\,\mu m$, the mean values are only slightly increased to mean $r_{eff,AaM} = 11.48\,\mu m$
and mean $r_{eff,M} = 10.60\,\mu m$.

Figure 14(e) illustrates a scatter plot of $\tau_M$ and $\tau_{AaM}$ sampled over all available overcast (black
dots) and partially cloudy (grey dots) pixels. There is a good agreement between the results from both
instruments, with most observations close to the 1:1 line and a correlation coefficient of $R = 0.992$
($R = 0.979$ for overcast pixels and $R = 0.968$ for partially cloudy pixels). The concentration of data
points around $0 > \tau_{AaM}, \tau_M < 5$ illustrates that partially cloudy pixels are characterized by very
low optical thicknesses. A scatter plot of $r_{eff,M}$ and $r_{eff,AaM}$ is shown in Figure 14(f). While there
is a good agreement for overcast pixels, illustrated by $R = 0.972$, there are visible deviations for
partially cloudy pixels ($R = 0.739$). As shown in Figure 14(e), most partially cloudy pixels exhibit
$\tau_M, \tau_{AaM} < 5$. In this part of the LUT the retrieval sensitivity is very low and even small uncertain-
ties in $\gamma_M$ and $\gamma_{AaM}$ yield large retrieval uncertainties for both $r_{eff,M}$ and $r_{eff,AaM}$. The fact that the





PDFs of $r_{\mathrm{eff,M}}$ and $r_{\mathrm{eff,AaM}}$ agree well shows that there is no preferred sign in the deviations (i.e., there is no overall overestimation or underestimation by one instrument). This implies that the relation between both retrieval products mostly resembles noise, indicating that retrieval uncertainties are the cause for the discrepancies. Including observations from partially cloudy pixels reduces the correlation coefficient for all $r_{\mathrm{eff,M}}$ and $r_{\mathrm{eff,AaM}}$ to $R = 0.851$. The slight slopes in the reflectance

relations yields slight slopes in the cloud property relations. For overcast pixels the derived slope values are $\alpha_\tau = 1.016$ (for the cloud optical thickness) and $\alpha_{\mathrm{r}} = 1.091$ (for the effective droplet radius), while the offset values are $\beta_\tau = -1.069$ and $\beta_{\mathrm{r}} = -1.061$. This implies a slight underestimation (overestimation) of the ASTER retrievals on the low (high) end of the respective value ranges.

**5.4 Uncertainty Contributions**

The analysis in Section 5.2.1–5.3.3 reveals a high agreement between the operational MODIS cloud retrieval products and the co–located ASTER results. This can be attributed to the use of the MODIS C6 retrieval algorithms and radiative transfer codes. Still, remaining uncertainties lead to the small differences in the cloud variable comparison. Besides the radiometric uncertainties of each instru-

ment, a number of factors impact the comparison between the MODIS and ASTER results.

Differences in the center wavelengths and SRF between the ASTER and MODIS bands, while theoretically accounted for in the applied radiative transfer codes, yield a remaining uncertainty not only in the reflectance comparison, but also in the retrieved cloud top, optical and microphysical properties. While the transmittance tables used in the retrieval algorithm of cloud top properties

are calculated for the full ASTER SRFs, the operational MODIS IR window retrieval and optimal estimation method are applied to the ASTER IR observations without any threshold adjustments. Spectral differences also impact the atmospheric correction algorithm. Since the land surface albedo product is created for MODIS bands 1–7 and there is no specific surface albedo product for ASTER, the SRF differences between ASTER and MODIS bands induce uncertainties in the derived spectral

surface albedo values. This is acknowledged by an increase in surface albedo uncertainty from 15% to 30% in the pixel–level uncertainty calculations. However, since the focus of this study is on MBL clouds sampled over ocean, this effect is mitigated by the use of ocean surface reflectances derived from the Cox–Munk model. For the reflectance comparison, the scale factors $f_{0.86,\mathrm{LUT}}$ and $f_{2.1,\mathrm{LUT}}$ theoretically provide the means to compare $\widehat{\gamma}_{0.86,\mathrm{M}}$ and $\widehat{\gamma}_{0.86,\mathrm{AaM}}$, as well as $\widehat{\gamma}_{2.1,\mathrm{M}}$ and $\widehat{\gamma}_{2.1,\mathrm{AaM}}$.

However, both are derived by means of radiative transfer simulations and are thus impacted by the involved assumptions (e.g., the ocean surface albedo, which might be different to the actually observed albedo). As mentioned in Section 3.3, derived $\tau_{\mathrm{M}}$ and $\tau_{\mathrm{AaM}}$ are scaled to MODIS band 1 and ASTER band 2, respectively (both centered around $0.65\,\mu\mathrm{m}$). This means that the comparison between $\tau_{\mathrm{M}}$ and $\tau_{\mathrm{AaM}}$ is also influenced by the different center wavelengths and SRF of this band.

These differences are in the range of the applied VNIR band (centered around $0.86\,\mu\mathrm{m}$). The band





differences also result in different vertical weighting functions (Platnick, 2000), which describe the vertical photon transport within the cloud and impact the retrieval of $r_{\mathrm{eff,M}}$ and $r_{\mathrm{eff,AaM}}$.

The aggregation of digital ASTER counts $d_{\mathrm{A}}(\Delta\lambda)$ within a $(1000 \cdot 1000)\,\mathrm{m}$ MODIS pixel is described in Section 5.1 and Figure 9. The co–location of ASTER and MODIS samples benefits from the small horizontal resolution of the ASTER measurements, the position of both instruments aboard Terra and an almost identical alignment of the respective pixels (i.e., pixel edges are almost parallel). Still, small co–location uncertainties remain. Here, a significant contribution comes from the full aggregation of $d_{\mathrm{A}}(\Delta\lambda)$ of ASTER pixels that are only partially within a MODIS pixel (right at the MODIS pixel edges). The resulting uncertainties have been derived for a number of example MODIS pixels, where the $d_{\mathrm{A}}(\Delta\lambda)$ values of partially included ASTER pixels have been weighted according to the respective area within the MODIS pixel. These computationally–expensive calculations reveal an uncertainty in the aggregated digital counts of $< 0.05$. However, the effect of this uncertainty is mitigated by the fact that the ASTER signal is stored as 8–bit unsigned integer values. Thus, aggregated ASTER counts are rounded to full integer values and cover a possible range of $0 - 255$. The resulting rounding error yields uncertainties in the derived reflectances $\gamma_{0.86,\mathrm{AaM}}$ and $\gamma_{2.1,\mathrm{AaM}}$, which get higher when the signal gets darker. For the 48 scenes presented in this paper, the maximum reflectance uncertainty introduced by this rounding error is $5\%$, associated with cloudy pixels characterized by $\gamma_{0.86,\mathrm{AaM}} \approx 0.03$. Moreover, uncertainties arise due to the MODIS point spread function (PSF), which characterizes the signal distribution within and outside a MODIS pixel (Huang et al., 2002). While $d_{\mathrm{A}}(\Delta\lambda)$ from each ASTER sample contributes equally to the aggregated signal, the MODIS PSF implies that the largest contribution in a MODIS signal comes from the center of the pixel, while there is also a noticeable influence from surrounding pixels.

Differences in $\theta_{\mathrm{s}}$ between ASTER and MODIS, which are $\Delta\theta_{\mathrm{s}} < 0.8°$ and $\Delta\theta_{\mathrm{s}} < 0.2°$ for the two example cases C14 and C19, respectively, can result in significant differences in the retrieved cloud variables. Especially around the cloud–bow and glory region uncertainties in the scattering angle can have a large impact on the sampled reflectances from both sensors. Moreover, for clouds with highly heterogeneous cloud–tops small differences in $\theta_{\mathrm{s}}$ imply that different parts of the cloud are sampled by each instrument.

Electronic crosstalk, which causes visible striping in the MODIS cloud property retrievals shown in Figure 8(b) and 8(d), induces additional uncertainties when comparing the operational MODIS and co–located ASTER results.

The ASTER cloud property retrieval at $15\,\mathrm{m}$ horizontal resolution requires that SWIR reflectances are scaled up to match the resolution of the VNIR band 3N. To estimate the retrieval uncertainties it is assumed that the variability of four SWIR reflectance samples at $30\,\mathrm{m}$ resolution within a $(60 \cdot 60)\,\mathrm{m}$ pixel is similar to the variability of four SWIR reflectance samples at $15\,\mathrm{m}$ resolution within a $(30 \cdot 30)\,\mathrm{m}$ pixel. For the data set presented in this study statistics about the difference $\Delta\gamma_{2.1,30-60}$ between actually observed SWIR reflectances at $30\,\mathrm{m}$ resolution and replicated values from the





60 m samples were derived. The median $\Delta\gamma_{2.1,30-60}$ from over 4.3 million pixels is naturally $0\%$. The $10^{\text{th}}$ and $90^{\text{th}}$, as well as the $25^{\text{th}}$ and $75^{\text{th}}$, percentile of $\Delta\gamma_{2.1,30-60}$ is $\pm 3.1\%$ and $\pm 1.3\%$,

respectively. This means that for most observations the uncertainties in retrieved $\tau_A$ and $r_{\text{eff,A}}$ due to the replication of SWIR reflectances at the highest ASTER resolution are estimated to be less than $\pm 0.5$ and $\pm 0.7\,\mu\text{m}$, respectively. However, since the comparison between MODIS and co–located ASTER results is done with aggregated digital counts and not the 15 m data, it is not affected by the replication of ASTER SWIR observations.

## 6   Summary and Conclusions

This study presents MODIS–like cloud property retrievals of MBL cloud optical and microphysical properties from high spatial resolution observations of the ASTER instrument aboard Terra. The ASTER retrievals of $\tau_A$ and $r_{\text{eff,A}}$, with a horizontal resolution as low as 15 m, are enabled by a research–level retrieval algorithm, which utilizes the operational MODIS C6 algorithm core.

The first objective of this paper is to document the retrieval scheme. The retrieval is based on the bispectral retrieval approach with pre–calculated LUTs and sampeld reflectances at ASTER bands 3N (centered around $\lambda = 0.810\,\mu\text{m}$ in the VNIR) and 5 (centered around $\lambda = 2.165\,\mu\text{m}$ in the SWIR). Because the central wavelengths and spectral response functions of the ASTER bands differ from the respective MODIS bands, ASTER–specific LUTs are applied in the cloud property

retrieval. Compared to the MODIS LUT the SWIR reflectances of ASTER are about $10\%$ larger, depending on the scene geometry. Since ASTER also lacks certain bands necessary for the MODIS cloud masking scheme, a new algorithm is introduced. It is based on five cloudiness thresholds and tested with about 210 ASTER MBL scenes ranging from homogeneous altocumulus to heterogeneous broken cloud fields. This data set also includes the cases presented in Zhao and Di Girolamo

(2006), where cloud amount is determined by an individual, single–band threshold for each scene. Comparisons between derived scene cloud covers from this single–band threshold and the new cloud masking algorithm show a high agreement with a median difference of about $0.4\%$. It is shown that pixels containing very thin clouds are potentially missed by the algorithm. However, only $0.03\%$ of these pixels are characterized by $\tau_A \geq 5$. The LUT collapses fast for $\tau_A < 5$, which significantly re-

duces the retrieval sensitivity and increases the uncertainties in the derived $\tau_A$ and $r_{\text{eff,A}}$. Examples of high–resolution ASTER retrievals are presented for two MBL scenes with different degrees of horizontal cloud heterogeneity. These cases demonstrate that the ASTER observations can resolve small scale, highly heterogeneous cloud structures, which are significantly smoothed by the MODIS measurements.

The second objective of this study is to compare co–located ASTER retrievals to the operational MODIS C6 results. The data set is provided by 48 MBL scenes sampled off the Coast of California. To match the MODIS sampling geometry, the digital ASTER counts at the original $(15 - 90)\,\text{m}$





horizontal resolution are aggregated within the respective MODIS pixels. The ASTER retrieval algorithm subsequently provides co–located ASTER results of $\widehat{\gamma}_{0.86,\mathrm{AaM}}$, $\widehat{\gamma}_{2.1,\mathrm{AaM}}$, $\tau_{\mathrm{AaM}}$ and $r_{\mathrm{eff,AaM}}$.

Moreover, the ASTER cloud mask at $15\,\mathrm{m}$ horizontal resolution yields a subpixel cloud cover for each aggregated pixel, which is used to discriminate between overcast and partially cloudy pixels. The data set amounts to $52,254$ overcast and $47,538$ partially cloudy pixels, where both ASTER and MODIS contain successful liquid water cloud property retrievals. PDFs and scatter plots of $\widehat{\gamma}_{0.86,\mathrm{AaM}}$, $\widehat{\gamma}_{2.1,\mathrm{AaM}}$, $\tau_{\mathrm{AaM}}$, and $r_{\mathrm{eff,AaM}}$ for both overcast and partially cloudy pixels agree well

with their MODIS counterparts, with similar value ranges and mean values. Highly positive correlations between sampled reflectances in the VNIR and SWIR, as well as between $\tau_{\mathrm{M}}$ and $\tau_{\mathrm{AaM}}$, are observed with Pearson's product–moment correlation coefficients $R > 0.980$. Correlations between $r_{\mathrm{eff,M}}$ and $r_{\mathrm{eff,AaM}}$ are lower with $R = 0.851$, primarily caused by larger differences for partially cloudy pixels. These deviations seem to be retrieval noise caused by increased retrieval uncertainties

due to the shape of the LUT for small $\tau_{\mathrm{M}}$ and $\tau_{\mathrm{AaM}}$. Limiting the data to only overcast pixels yields $R = 0.972$ for the effective droplet radius comparison. However, slight slopes in the reflectance relations yield similar slopes in the cloud product relations, indicating an overestimation of the ASTER results for small values and an underestimation of the ASTER results for larger values. The overall good agreement between the MODIS and ASTER retrievals is confirmed for two example cases.

While the rather homogeneous case C14 is characterized by deviations of $\pm1.82$ and $\pm1.09\,\mu\mathrm{m}$ in retrieved cloud optical thickness and effective droplet radius, respectively, partially cloudy pixels sampled for the inhomogeneous case C19 are characterized by differences between $r_{\mathrm{eff,AaM}}$ and $r_{\mathrm{eff,M}}$ of up to $\pm12\,\mu\mathrm{m}$. Several uncertainty factors in the ASTER and MODIS cloud property comparison are presented.

Not discussed in this study are the comparison of cloud phase and cloud top height retrievals. Since the focus of this study is on MBL scenes over the ocean and the 48 scenes were selected to contain no overlying cirrus, $99\%$ of all cloudy pixels are characterized to contain liquid water clouds by both ASTER and MODIS. Frequency distributions and statistics of the cloud top height comparison show a high agreement between both instruments. Mean cloud top heights of $670\,\mathrm{m}$ and

$823\,\mathrm{m}$ are observed from ASTER and MODIS, respectively. However, the MODIS cloud top height retrieval is performed at a horizontal resolution of $5\,\mathrm{km}$, averaging observations from a $5\cdot5$ pixel array with a horizontal resolution of $1\,\mathrm{km}$. This means that only $8,037$ MODIS pixels are included in the analysis (compared to over $150,000$ ASTER pixels). Further studies with a more comprehensive data set, consisting of different cloud types in different altitudes and with different thermodynamical

phases, is required to make a statement about the reliability of the ASTER cloud phase and cloud top height retrieval.

This paper illustrates that the research–level retrieval algorithm for ASTER observations yields reliable cloud property retrievals comparable to the operational MODIS C6 results. The unique ASTER retrievals will enable a number of interesting future studies. ASTER observations at native resolu-





tion can help in determining the subpixel cloud structure of partially cloudy pixels, which result in large uncertainties in the cloud property retrieval. This can improve the understanding of MODIS PCL retrievals and their uncertainties. Moreover, the ASTER observations at the native resolutions can be aggregated to ever larger horizontal resolutions. Together with the knowledge about the true subpixel cloud structure, such scale–analysis studies will provide valuable insights into 3D radiative

effects and the impacts of resolved and unresolved variability in cloud remote sensing.

*Acknowledgements.* This study is supported by NASA grants NNX14AJ25G and NNX15AC77G. The hardware used in the computational studies is part of the UMBC High Performance Computing Facility (HPCF). The facility is supported by the U.S. National Science Foundation through the MRI program (grant nos. CNS-0821258 and CNS-1228778) and the SCREMS program (grant no. DMS-0821311), with additional substantial

support from the University of Maryland, Baltimore County (UMBC). The MODIS data are obtained from NASA's Level 1 and Atmosphere Archive and Distribution System (LAADS, http://ladsweb.nascom.nasa.gov/). ASTER data are obtained by the EarthExplorer interface (http://earthexplorer.usgs.gov) provided by the United States Geological Survey (USGS).





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





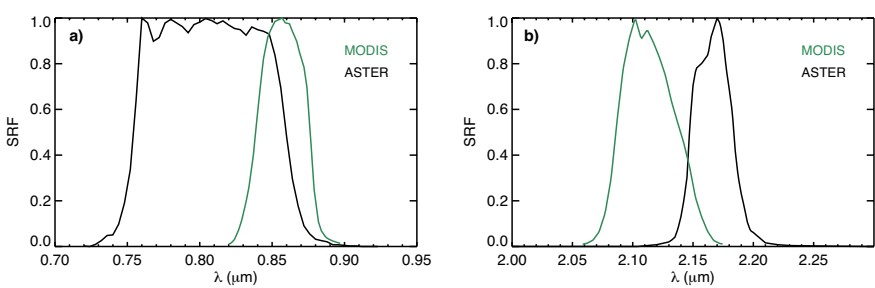

**Figure 1.** (a) Spectral response function (SRF) of the VNIR band signal for MODIS (green) and ASTER (black) as a function of wavelength ($\lambda$). (b) Same as (a) but for the signal in the SWIR band.





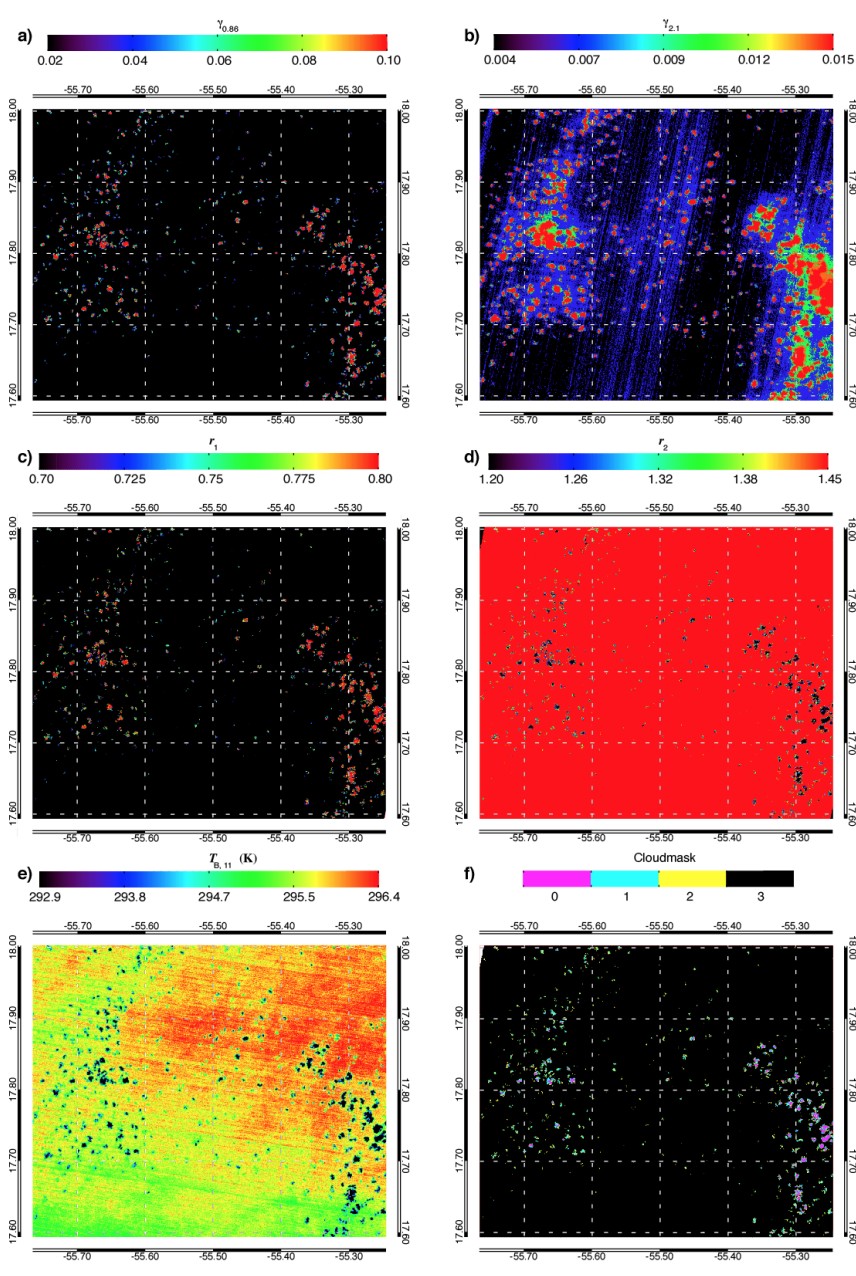

**Figure 2.** (a) Band 3N reflectance ($\gamma_{0.86}$) from ASTER observations on 01/22/2005 (i.e., cloud mask test (i)). (b) Same as (a) but showing the band 5 reflectance ($\gamma_{2.1}$) (i.e., cloud mask test (ii)). (c) Same as (a) but showing the color ratio $r_1$ (i.e., cloud mask test (iii)). (d) Same as (a) but showing the color ratio $r_2$ (i.e., cloud mask test (iv)). (e) Same as (a) but showing the brightness temperature $T_{B,11}$ (i.e., cloud mask test (v)). (f) Same as (a) but showing the cloudiness flags '0'–'3', after applying cloud mask tests (i)–(v).



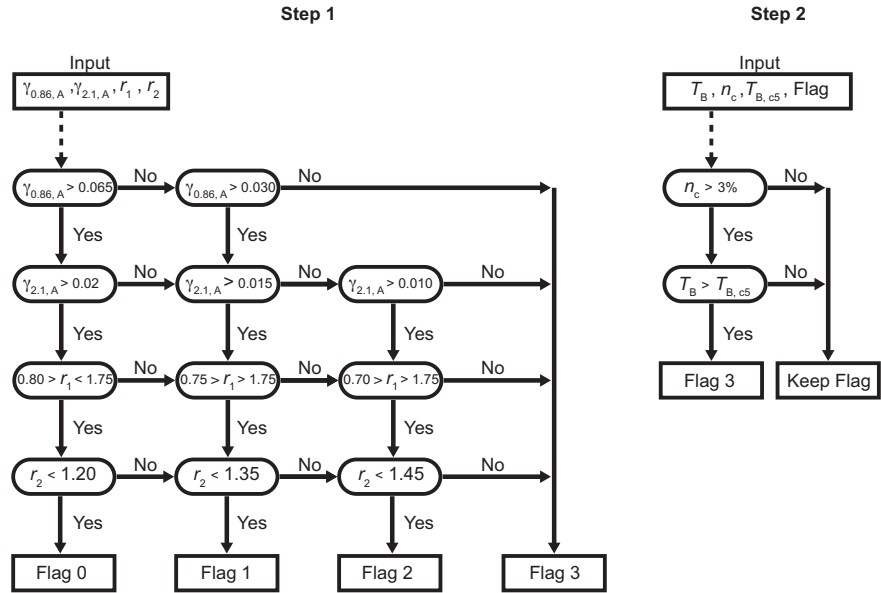

**Figure 3.** Outline of the new ASTER cloud mask algorithm. Step 1 illustrates the decision–tree including cloudiness tests (i)–(iv) based on ASTER band 3N and 5 reflectances $\gamma_{0.86}$ and $\gamma_{2.1}$, as well as color ratios $r_1$ and $r_2$. Step 2 illustrates the correction for complex broken cumulus scenes, as well as cases with pronounced sun glint. This correction, test (v), is based on the derived cloudiness flags from step 1, the brightness tempera­ture $T_{B,11}$ (calculated from the ASTER band 14 radiances), the percentage of clear pixels with cloudiness flag '3' ($n_c$), and the $5^{\text{th}}$ percentile of $T_{B,11}$ sampled over all clear pixels ($T_{B,c5}$).





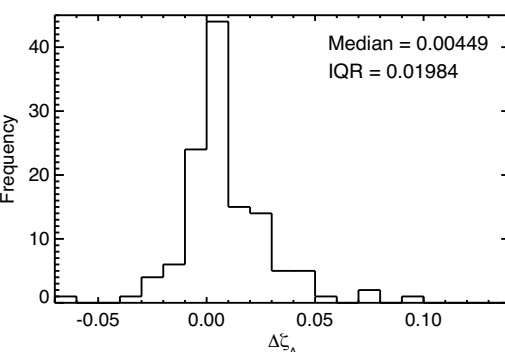

**Figure 4.** Difference in domain–averaged cloud cover ($\Delta C_A$) between the single–band threshold reported by Zhao and Di Girolamo (2006) and the new ASTER cloud masking scheme. Data was sampled over 124 broken cumulus scenes in the tropical, southern Atlantic Ocean. Values for the median and interquartile range (IQR) are given.





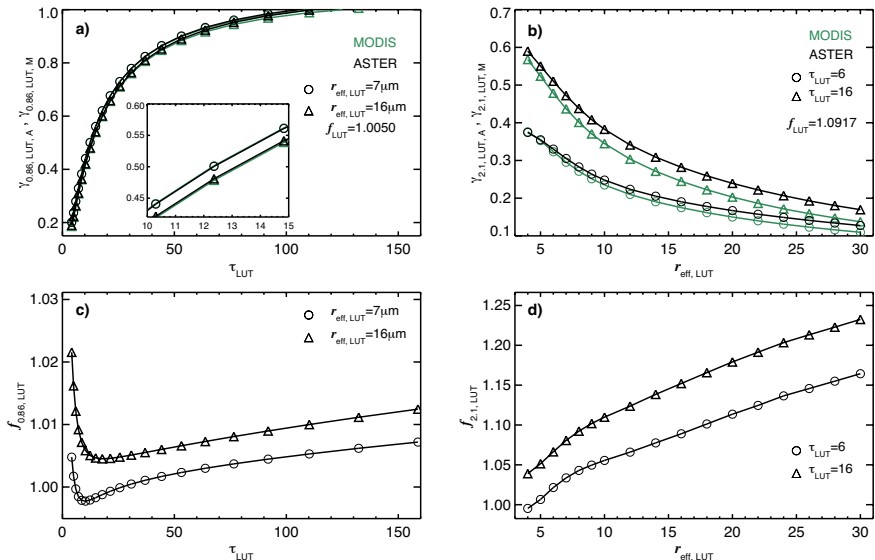

**Figure 5.** (a) VNIR reflectance from the respective lookup tables (LUT) for ASTER ($\gamma_{0.86,\mathrm{LUT,A}}$, black) and MODIS ($\gamma_{0.86,\mathrm{LUT,M}}$, green) as a function of input cloud optical thickness ($\tau_{\mathrm{LUT}}$). The input solar zenith angle is $\theta_0 = 24°$, the solar azimuth angle is $\varphi_0 = 143°$, and the sensor zenith angle is $\theta_s = 8.59°$. Symbols indicate a fixed input cloud effective radius ($r_{\mathrm{eff,LUT}}$) of $r_{\mathrm{eff,LUT}} = 7\,\mu\mathrm{m}$ (open circles) and $r_{\mathrm{eff,LUT}} = 16\,\mu\mathrm{m}$ (open triangles). The inlay shows a close–up of the region between $\tau_{\mathrm{LUT}} = 10 - 15$. (b) SWIR reflectance from the respective LUT $\gamma_{2.1,\mathrm{LUT,A}}$ (ASTER) and $\gamma_{2.1,\mathrm{LUT,M}}$ (MODIS) as a function of input $r_{\mathrm{eff,LUT}}$. Symbols indicate fixed $\tau_{\mathrm{LUT}} = 6$ (open circles) and $\tau_{\mathrm{LUT}} = 18$ (open triangles). (c) The theoretical scale factor $f_{0.86,\mathrm{LUT}}$ between $\gamma_{0.86,\mathrm{LUT,A}}$ and $\gamma_{0.86,\mathrm{LUT,M}}$ as a function of $\tau_{\mathrm{LUT}}$. (d) The theoretical scale factor $f_{2.1,\mathrm{LUT}}$ between $\gamma_{2.1,\mathrm{LUT,A}}$ and $\gamma_{2.1,\mathrm{LUT,M}}$ as a function of $r_{\mathrm{eff,LUT}}$.



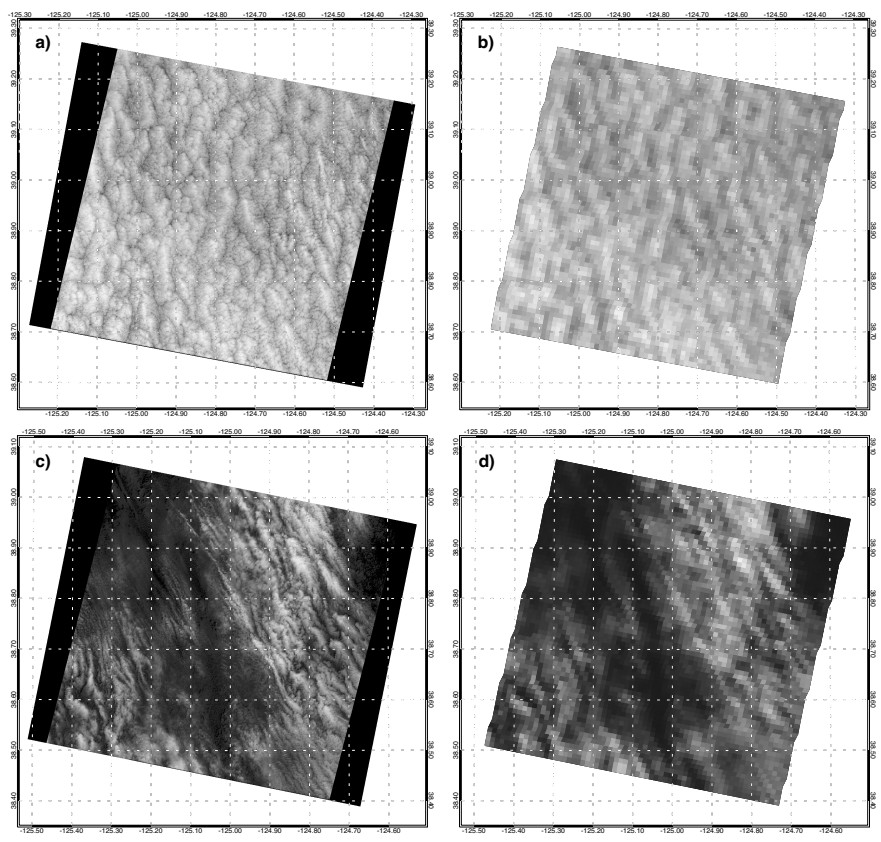

**Figure 6.** (a) Single–band grayscale image of band 3N reflectances sampled by ASTER on 05/13/2003 off the Coast of California (scene C14). (b) Same as (a) but from band 2 reflectances sampled by MODIS. (c)–(d) Same as (a)–(b) but sampled on 06/10/2005 (scene C19).





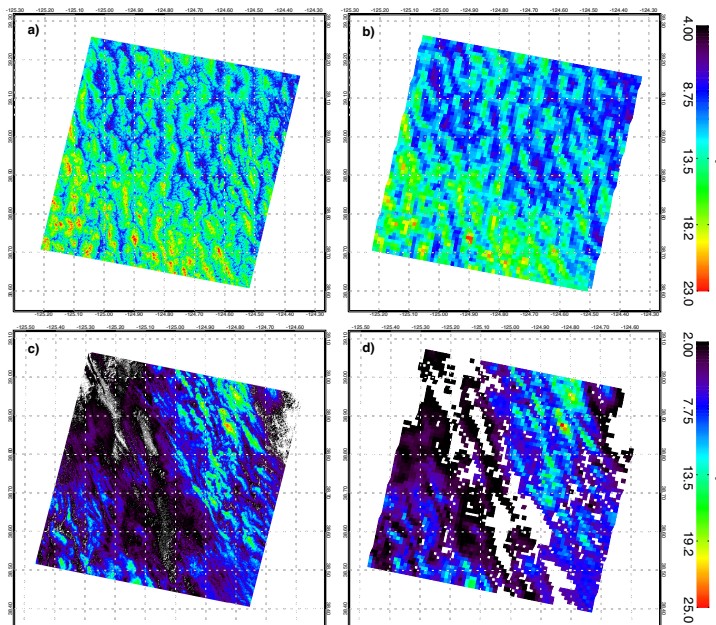

**Figure 7.** (a) Map of retrieved cloud optical thickness from reflectances sampled by ASTER ($\tau_A$) on 05/13/2003 (C14). (b) Same as (a) but showing the operational MODIS retrieved cloud optical thickness ($\tau_M$). (c)–(d) Same as (a)–(b) but for observations on 06/10/2005 (C19).





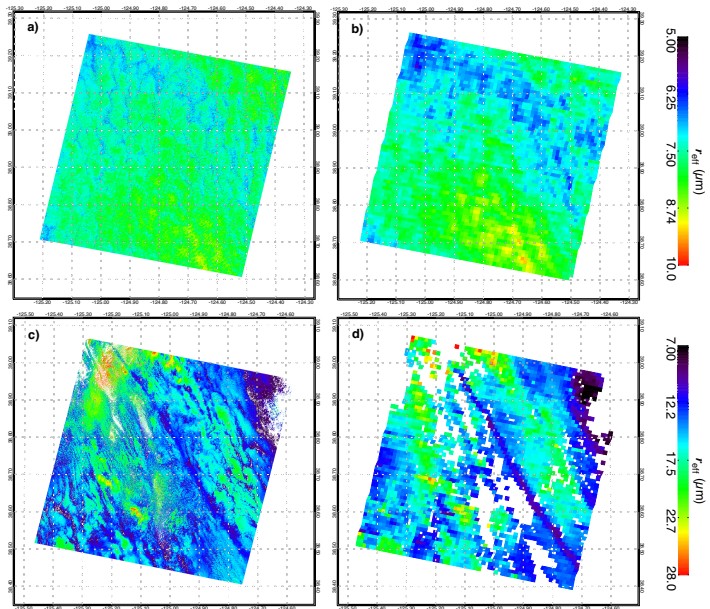

**Figure 8.** (a) Map of retrieved effective droplet radius from reflectances sampled by ASTER ($r_{\mathrm{eff,A}}$) on 05/13/2003 (C14). (b) Same as (a) but showing the operational MODIS retrieved effective droplet radius ($r_{\mathrm{eff,M}}$). (c)–(d) Same as (a)–(b) but for observations on 06/10/2005 (C19).





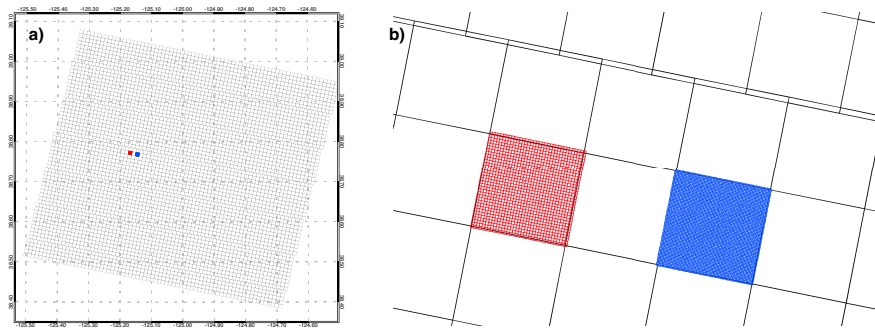

**Figure 9.** (a) Derived MODIS pixels (grey lines) for the MBL scene observed on 06/10/2005 (C19). For two individual MODIS pixels all co–located ASTER pixels in the VNIR (blue lines) and SWIR (red lines) are shown, wich are characterized by a horizontal resolution of 15 m and 30 m, respectively. (b) Close–up of the two MODIS pixels.




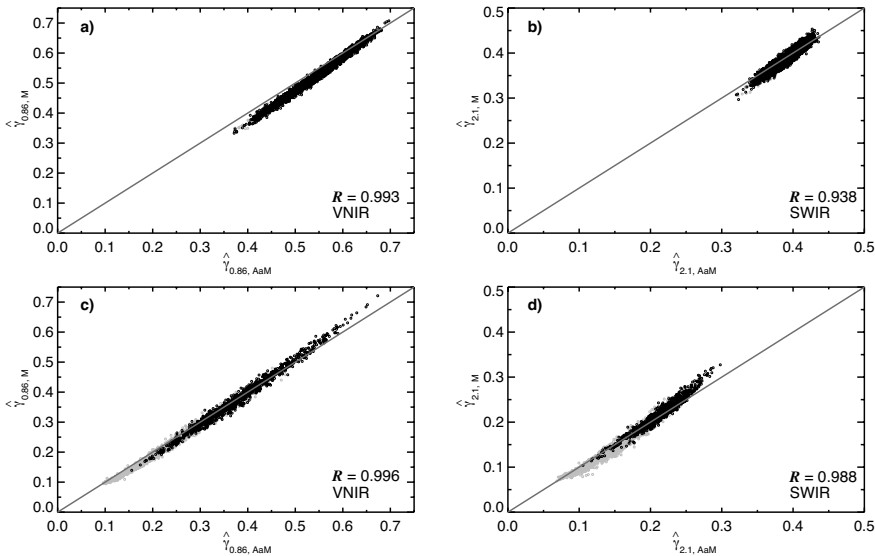

**Figure 10.** (a) Scatter plot of atmospherically corrected MODIS reflectances ($\widehat{\gamma}_{0.86,\mathrm{M}}$) in the VNIR as a function of co–located, atmospherically corrected ASTER reflectances ($\widehat{\gamma}_{0.86,\mathrm{AaM}}$) in the VNIR. The gray diagonal line represents the 1:1 line. Overcast (partially cloudy) pixels are indicated in black (grey) color. Data is from observations on 05/13/2003 (C14). (b) Same as (a) but for $\widehat{\gamma}_{2.1,\mathrm{M}}$ and $\widehat{\gamma}_{2.1,\mathrm{AaM}}$ sampled in the SWIR. (c)–(d) Same as (a)–(b) but for observations on 06/10/2005 (C19).




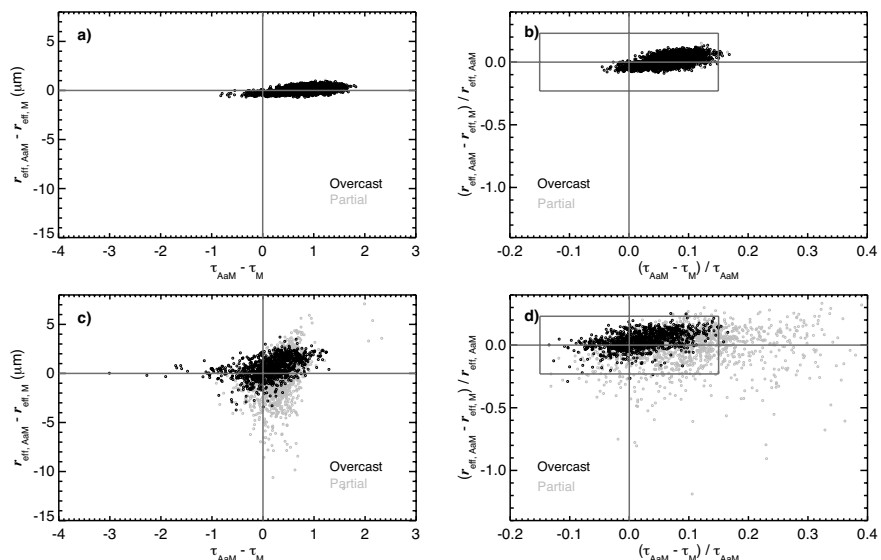

**Figure 11.** (a) Difference between effective droplet radius retrieved from co–located ASTER observations ($r_{\mathrm{eff,AaM}}$) and the operational MODIS product ($r_{\mathrm{eff,M}}$) versus difference between cloud optical thickness retrieved from co–located ASTER observations ($\tau_{\mathrm{AaM}}$) and the operational MODIS product ($\tau_{\mathrm{M}}$). Only data points, where both ASTER and MODIS retrievals have a successful liquid water cloud retrieval, are shown. Colors indicate samples over overcast (black) and partially cloudy pixels (grey). The gray horizontal and vertical lines indicate the points where no deviation between ASTER and MODIS retrievals occur. Data is from observations on 05/13/2003 (C14). (b) Same as (a) but normalized by $r_{\mathrm{eff,AaM}}$ and $\tau_{\mathrm{AaM}}$, respectively. The gray box indicates the mean retrieval uncertainty for $r_{\mathrm{eff,AaM}}$ and $\tau_{\mathrm{AaM}}$, calculated by applying the absolute radiometric uncertainties of ASTER band 3N and 5 reflectances. (c)–(d) Same as (a)–(b) but for observations on 06/10/2005 (C19).





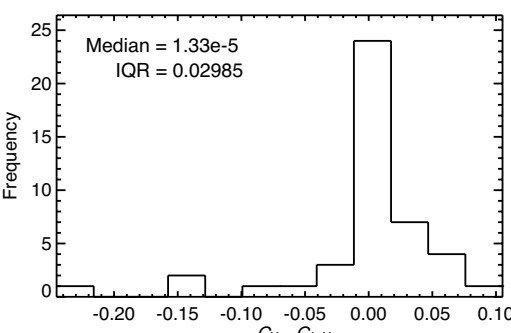

**Figure 12.** Difference in domain–averaged cloud cover based on the operational MODIS cloud mask ($C_M$) and co–located ASTER observations ($C_{AaM}$). Values for the median and interquartile range (IQR) are given.

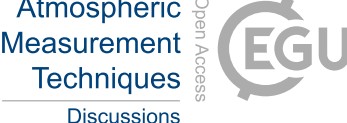



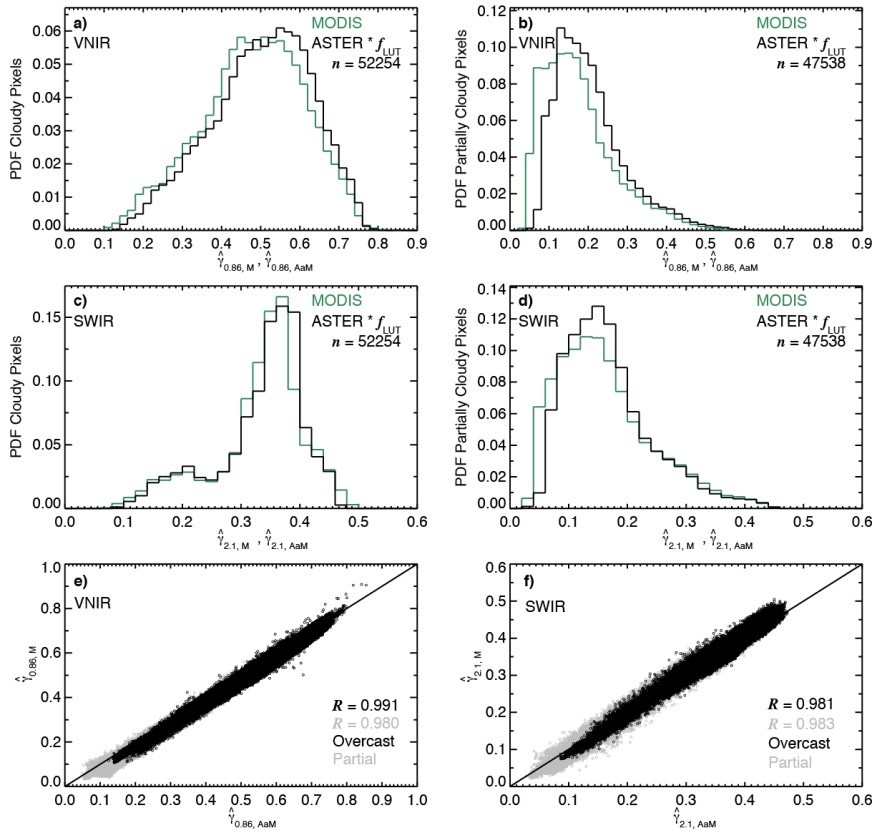

**Figure 13.** (a) PDFs of atmospherically corrected MODIS VNIR reflectances ($\widehat{\gamma}_{0.86,\mathrm{M}}$) and co–located, atmospherically corrected ASTER VNIR reflectances ($\widehat{\gamma}_{0.86,\mathrm{AaM}}$), multiplied by the derived scale factor between ASTER and MODIS LUT reflectances ($f_{0.86,\mathrm{LUT}}$, see Section 2.3). Only overcast pixels, containing both a successful MODIS and ASTER liquid water cloud retrieval, from the 48 MBL cases are considered in the calculation of the PDFs. The number of samples ($n$) is given. (b) Same as (a) but for partially cloudy pixels. (c)–(d) Same as (a)–(b) but for $\widehat{\gamma}_{2.1,\mathrm{M}}$ and $\widehat{\gamma}_{2.1,\mathrm{AaM}}$ sampled in the SWIR. (e) Scatter plot of $\widehat{\gamma}_{0.86,\mathrm{M}}$ and $\widehat{\gamma}_{0.86,\mathrm{AaM}}$ (multiplied by $f_{2.1,\mathrm{LUT}}$) in the VNIR. Overcast (partially cloudy) pixels are indicated in black (grey) color. The diagonal line represents the 1:1 line. (f) Same as (e) but for reflectances in the SWIR.





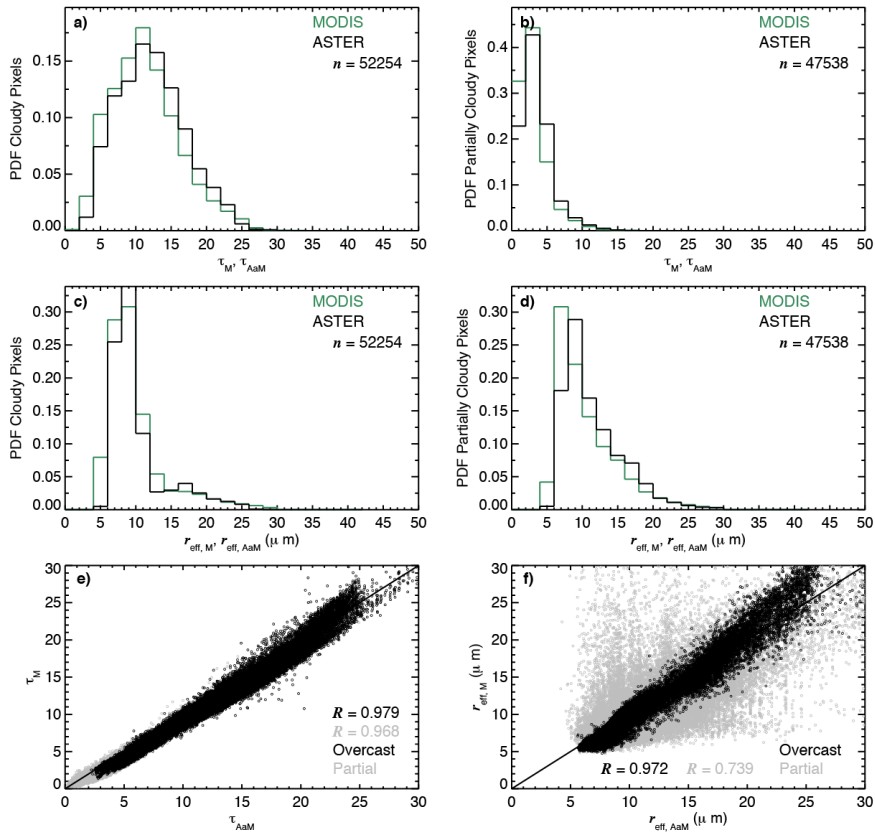

**Figure 14.** (a) PDFs of cloud optical thickness retrieved from MODIS reflectances ($\tau_M$) and co–located ASTER reflectances ($\tau_{AaM}$). Only overcast pixels, containing both a successful MODIS and ASTER liquid water cloud retrieval, from the 48 MBL cases are considered in the calculation of the PDFs. The number of samples ($n$) is given. (b) Same as (a) but for partially cloudy pixels. (c)–(d) Same as (a)–(b) but for the effective droplet radius retrieved from MODIS reflectances ($r_{eff,M}$) and co–located ASTER reflectances ($r_{eff,AaM}$). (e) Scatter plot of $\tau_M$ and $\tau_{AaM}$. Overcast (partially cloudy) pixels are indicated in black (grey) color. The diagonal line represents the 1:1 line. (f) Same as (e) but for $r_{eff,M}$ and $r_{eff,AaM}$.





**Table 1.** Overview of ASTER band numbers, wavelength range ($\Delta\lambda$) covered by each band, spatial resolution ($\Delta x$), as well as unit conversion coefficients ($C_H$, $C_N$, $C_{L1}$, $C_{L2}$) for High, Normal, Low1 and Low2 gains, respectively.

| Band | $\Delta\lambda$ ($\mu$m) | $\Delta x$ (m) | $C_H$ $\mathrm{W\,m^{-2}\,\mu m^{-1}\,sr}$ | $C_N$ $\mathrm{W\,m^{-2}\,\mu m^{-1}\,sr}$ | $C_{L1}$ $\mathrm{W\,m^{-2}\,\mu m^{-1}\,sr}$ | $C_{L2}$ $\mathrm{W\,m^{-2}\,\mu m^{-1}\,sr}$ |
|---|---|---|---|---|---|---|
| 1 | 0.520-0.600 | 15 | 0.676 | 1.688 | 2.25 | - |
| 2 | 0.630-0.690 | 15 | 0.708 | 1.415 | 1.89 | - |
| 3N | 0.760-0.860 | 15 | 0.423 | 0.862 | 1.15 | - |
| 3B | 0.760-0.860 | 15 | 0.423 | 0.862 | 1.15 | - |
| 4 | 1.600-1.700 | 30 | 0.1087 | 0.2174 | 0.290 | 0.290 |
| 5 | 2.145-2.185 | 30 | 0.0348 | 0.0696 | 0.0925 | 0.409 |
| 6 | 2.185-2.225 | 30 | 0.0313 | 0.0625 | 0.0830 | 0.390 |
| 7 | 2.235-2.285 | 30 | 0.0299 | 0.0597 | 0.0795 | 0.332 |
| 8 | 2.295-2.365 | 30 | 0.0209 | 0.0417 | 0.0556 | 0.245 |
| 9 | 2.360-2.430 | 30 | 0.0159 | 0.0318 | 0.0424 | 0.265 |
| 10 | 8.125-8.475 | 90 | - | 0.006822 | - | - |
| 11 | 8.475-8.825 | 90 | - | 0.006780 | - | - |
| 12 | 8.925-9.275 | 90 | - | 0.006590 | - | - |
| 13 | 10.250-10.950 | 90 | - | 0.005693 | - | - |
| 14 | 10.950-11.650 | 90 | - | 0.005225 | - | - |





**Table 2.** Case number (C1–C48) and sample date of each MBL scene in this study. The date format is MM/DD/YYYY Hour:Minute:Second.

| Case number # | Date | Case number # | Date | Case number # | Date |
|---|---|---|---|---|---|
| 1 | 03/02/2006/ 19:14:44 | 21 | 06/25/2004/ 19:10:45 | 41 | 10/06/2003/ 19:04:27 |
| 2 | 03/06/2005/ 19:20:37 | 22 | 07/04/2007/ 19:09:35 | 42 | 10/21/2006/ 19:09:31 |
| 3 | 03/06/2005/ 19:20:46 | 23 | 07/04/2007/ 19:10:19 | 43 | 10/25/2005/ 19:14:44 |
| 4 | 03/06/2005/ 19:20:55 | 24 | 07/04/2007/ 19:10:46 | 44 | 10/25/2006/ 18:45:26 |
| 5 | 03/06/2005/ 19:21:04 | 25 | 07/11/2007/ 19:16:06 | 45 | 10/25/2006/ 18:45:35 |
| 6 | 03/06/2005/ 19:21:13 | 26 | 07/20/2007/ 19:10:07 | 46 | 10/30/2006/ 19:03:35 |
| 7 | 03/08/2005/ 19:08:35 | 27 | 07/20/2007/ 19:10:16 | 47 | 12/03/2005/ 19:20:56 |
| 8 | 03/08/2005/ 19:08:44 | 28 | 07/20/2007/ 19:10:25 | 48 | 12/16/2004/ 19:20:41 |
| 9 | 03/08/2005/ 19:08:53 | 29 | 08/18/2006/ 19:09:01 | | |
| 10 | 04/19/2006/ 19:14:55 | 30 | 08/18/2006/ 19:09:18 | | |
| 11 | 04/19/2006/ 19:15:13 | 31 | 08/26/2003/ 19:09:37 | | |
| 12 | 04/19/2006/ 19:15:22 | 32 | 08/26/2003/ 19:09:55 | | |
| 13 | 04/19/2006/ 19:15:31 | 33 | 08/26/2003/ 19:10:12 | | |
| 14 | 05/13/2003/ 19:15:46 | 34 | 08/29/2006/ 18:52:02 | | |
| 15 | 05/30/2006/ 19:08:57 | 35 | 08/29/2006/ 18:52:11 | | |
| 16 | 06/02/2007/ 19:09:29 | 36 | 09/02/2003/ 19:15:12 | | |
| 17 | 06/02/2007/ 19:09:47 | 37 | 09/07/2005/ 19:14:31 | | |
| 18 | 06/03/2005/ 19:14:42 | 38 | 09/07/2005/ 19:14:49 | | |
| 19 | 06/10/2005/ 19:20:47 | 39 | 09/10/2006/ 19:15:21 | | |
| 20 | 06/10/2005/ 19:21:04 | 40 | 09/11/2004/ 19:21:08 | | |