# Peer review of "Marine boundary layer cloud property retrievals from high–resolution ASTER observations: Case studies and comparison with Terra–MODIS"

_Atmospheric Measurement Techniques, 2016_

## Referee Comment (RC1) · Anonymous Referee #1 · 29 Aug 2016

'Marine boundary layer cloud property retrievals from high-resolution ASTER observations: case studies and comparison with Terra-MODIS', F. Werner, G. Wind, Z. Zhang, S. Platnick, L. Di Girolamo, G. Zhao, N. Amarasinghe, and K. Meyer, submitted to AMT

This manuscript describes a long overdue attempt at cloud optical thickness (COT) and effective radius (CER) retrievals from the ASTER instrument using a MODIS-like look-up table approach. Some adjustments have to be made because of spectral response function differences but overall the approach is very similar. As ASTER and MODIS are located onboard EOS-Terra that makes pixel-level comparisons (as done in this paper)

straightforward with new insights gleaned on resolution-dependent issues between the two instruments.

This is an excellent paper that is very straightforward in its presentation and messaging with two primary purposes: to describe the retrieval, and compare the ASTER results to MODIS Collection 6 retrievals. A couple of case studies are presented in detail with images shown and details of the scene described at length, then 48 scenes are shown for statistical comparisons between ASTER and MODIS. In my opinion, there is little to improve upon in this paper and I only have a few minor revisions to suggest and comments to make below.

What about the land challenge? Can this approach be extended to land retrievals without much development? Are there plans to do so, and if not, why not?

p. 3 around lines 65-75: there is at least one other attempt at an ASTER cloud mask that could be considered for citation (if deemed relevant): Hulley, G. C., and S. J. Hook (2008), A new methodology for cloud detection and classification with ASTER data, Geophys. Res. Lett., 35, L16812, doi:10.1029/2008GL034644.

Lines 101-102: do the authors think that the backward-viewing direction has some potential to improve upon the ASTER COT and CER retrievals? If so, what geophysical field(s) would this benefit the most?

Section 3.2.1 cloud top properties: First, it wasn't fully clear how the ASTER pixel level data was averaged up to the MODIS resolution. Second, are all ASTER retrievals done at the ASTER resolution then averaged to the MODIS resolution for comparison, or are the reflectances/radiances averaged then the retrieval is performed? It appears that the former was the case for most, if not all, of the paper but it should be made clearer or in a few places if made already. Third, are retrievals compared if ASTER is partly cloudy within a cloud-identified MODIS pixel?

Line 413: constrained

Lines 580-587: how do the ASTER uncertainties fall within the MODIS uncertainties? Does this work yield further insight on the character of MODIS COT and CER pixel-level uncertainty estimates?

Lines 753-756: it is somewhat unclear in the way currently described how the ASTER uncertainties are obtained. Maybe a graphic, table, or improved discussion will help clarify.

Figure 3: the authors may want to consider labeling flag=0,1,2,3 also as the four categories of clear/cloudy and how they map to the MODIS cloud mask. Labeling as flag=0,1,2,3 makes the figure less useful because you have to page to the discussion to figure it out.

Figure 9: are MODIS pixels really perfect squares? How does the spatial weighting of the reflectance as seen by the instrument look like within the pixel? Same point about ASTER. Any references of previous work discussing current state of knowledge of pixel-level characteristics would benefit the methodology of the paper.

Figure 11: Impossible to see 'partial' in the upper row. May consider shrinking y-axis scale or presenting data in a different manner if seeing gray points is a key take-away from this figure.

---

## Referee Comment (RC2) · Anonymous Referee #2 · 16 Sep 2016

* Overall comment This paper is a technical report that discusses retrievals of cloud properties obtained from ASTER (new generating) and MODIS C6 (existing). The significance of this paper is that this paper reveals bias and differences between ASTER and MODIS products. However, discussion is too long comparing with the volume of contents and a little bit dull. I recommend to authors to refine this manuscript e.g. shorten the paragraphs and focus on a few topics that authors like to emphasis the most.

* Questions and minor correction Line 74: The sentence "there remains no cloud property retrieval algorithm for ASTR" contradicts to the sentence "(Line 66) Zhao and Di and Dey et al. use the high –resolution ASTER reflectance measurements at the lambda=0.86um band to derive a statistical description of the macrophysical properties of trade wind clouds". Please clarify this point.

Line 106 and Line 485, Line 723 etc.: I don't understand the meaning of symbol "delta" in dA(delta lambda).

Line 180: Is it reasonable to use the thresholds of 0.03 and 0.065 that obtained from MODIS algorithm to ASTER analysis?

In Fig 3 : 0.80>r1<1.75 seems an error. Please correct it.

Line 204: Is it reasonable to use the thresholds of 0.065, 0.02, 0.80, 1.75 that obtained from MODIS algorithm to ASTER analysis?

Line 215: How to obtain the values of 33.4egs, 63.2degs, and aerosol optical thickness of (0.04-1.49) ? Please clarify this point.

Lin 230: How to scale up the TB11? Please clarify this point.

Line 244: What does "a single case-by-case threshold for gamma0.86.A" mean?

Line 255: -0.04> delta CA < 0.04 seems an error. Please correct it.

Line 296: Why do you use such a complex method such as "If the calculated cloud top pressure is larger than 650mb the operational MODIS C6 IR window retrieval algorithm is used to calculate the final value of cloud top pressure" ? Please clarify this point.

Line 337 and Line 326: There are two similar sentences to explain the Cox and Munk (1954a, b). Could you unify these two sentences into one?

Line 348: How to scale up SWIR (30m) in to 15m spatial resolution?

Line 370- and Eq (4): I cannot understand significance of examining the scale factor f0.86m LUT, because you have set up ASTER LUT and MODIS LUT, individually. There

are no mean to examine ratio (scale factor) of LUT values. Please clarify this point.

Line 406: It is difficult to understand "124 ASTER scenes" and "48 MBL cloud case".

Line 412: 48 MBL cloud cases were sampled between 05/2003 and 07/2007. However, in line 123, it is mentioned that "the SWIR signal started to suffer from anomalous striping and saturation of values due to an increase in the SWIR detector starting in May 2007". Is it unreasonable to use 05/2007 to 07/2007 data for the cloud retrieval?

Line 416: How to eliminate overlying cirrus scenes and complex multi-layered cloud system?

Line 457: Why there appeared cloud holes in MODIS than ASTER?

Line 468: It is difficult to identify "visible striping in the reff, M results in this figure.

Line 497: what does 4400 and 1100 pixels mean in this sentence.

Line 549: It is difficult to identify black circles and gray circles in the Fig.11 because the figure size is too small. It may be a technical problem but significant to audiences. Please improve Fig. 11.

Line 573: I cannot find "-12.36um" of r_eff, AaM- r_eff,M in the figure 11.

Line 663: What does "Tau_M, Tau_AaM=0.17-24.92." mean? I cannot find that range of values of Tau between 0.17 to 24.92 in Fig. 14(b).

Line 668: What does "r_eff,M, r_eff, AaM = 4.76um"? I cannot find that 4.76um will have a specific values in Fig. 14 (c ).

Line 672: 10.00um>r_eff,M, r_eff,AaM<20.00um will be an error.

Line 679: 0>Tau_AaM, Tau_M< 5 will be an error.

Line 710: Where does "15% to 30%" come from. Can you add some references?

Line 749: I cannot identify visible striping in Fig 8(b) and 8(d).

That's all of my comments and questions.

---

## Referee Comment (RC3) · Anonymous Referee #3 · 16 Sep 2016

General

The paper addresses an important research question on the applicability / validity of 1D radiative transfer calculations for high spatial resolution cloud property retrievals. Thereto the authors adapt an existing retrieval scheme for large pixels from MODIS to small pixels from ASTER.

This paper fits well into AMT. The scale dependence of cloud property retrievals is very important question and the research is very relevant for interpretation of satellite cloud retrievals.

[Figure]

The paper is well-written, contains important results based on solid work, and will probably lead to future papers on the same topic. The number of figures is large, and could possibly be reduced. The paper can be accepted when the following comments are taken into account.

Main comments:

(1) The complication of atmospheric absorption in the wide VNIR band of ASTER (channel 3) is hardly discussed. Atmospheric correction has a much stronger effect for ASTER due to its broad VNIR band than for MODIS. In the broad VNIR channel of ASTER, the O2 A-band absorption and H2O band absorption play a large role. Please describe how you correct for these atmospheric absorption bands. The correction will depend on cloud height: the lower the cloud, the more correction is needed. Please show the sensitivity of the correction to cloud height. Please show the atmospheric absorption spectrum together with the spectral response function of the instrument bands in Figure 1.

(2) Can you already give conclusions on 3D effects seen in cloud retrievals from ASTER?

Minor and textual comments:

Abstract:

l. 8, l. 10, etc.: symbols with subscripts lead to too heavy notation. Please shorten where possible.

\gamma is a strange symbol for reflectance. Please use R or \rho.

l. 13: There are too many details in the abstract.

l. 20 ff: So is 1D retrieval good enough at these small scales? This is an important finding.

Are there biases due to 3D RT for fully cloud covered pixels ? The effect of broken

clouds on biases in reff is well known for 1D RT clouds. (E.g. Wolters et al., JGR, vol. 115, D10214, doi:10.1029/2009JD012205, 2010).

Section 1: At the end of the introduction, please briefly outline the setup of the paper.

l.101: in this spectral range oxygen and H2O absorption might be a problem

l. 118-120: These solar spectrum references are pretty old. Why not use a modern composite synthetic solar spectrum, like Gueymard (Solar Energy, 2004)?

l. 142: Please remove the brackets. This occurs at many places in paper.

l. 312: This correction will depend on cloud top height.

l. 386: what is the reason of this difference?

l. 396-398: what is the physical reason that the ASTER observation is brighter?

l. 704-705: Absorption by O2 and H2O in the VNIR band does not only affect above-cloud correction, but also the cloud reflectance itself due to multiple scattering and absorption inside the clouds.

Figure 2: what kind of scene is this ? what about cloudiness ? please also give the gray scale image of the scene.

Figure 5: Caption: is the solar azimuth also the viewing – solar azimuth difference, which is the relevant quantity? What is the scale factor?

Figure 6: how large is this scene?

Figure 7: please refer to the previous figure.

---

## Author Comment (AC1) · 1 Nov 2016

We'd like to thank the editor for handling our manuscript, as well as reviewer #1 for reading our manuscript and providing numerous, helpful comments. We have carefully read through all the comments and questions and revised the manuscript accordingly. Please find our point-to-point response to reviewer #1 below. Here, the reviewer's general remarks are formatted to be left-aligned text in italic font, the specific questions/comments are shown in left-aligned text in bold and italic font, while our responses are indented and formatted in regular font.

**Point-to-point Response to Anonymous Referee #1**

*'Marine boundary layer cloud property retrievals from high-resolution ASTER observations: case studies and comparison with Terra-MODIS', F. Werner, G. Wind, Z. Zhang, S. Platnick, L. Di Girolamo, G. Zhao, N. Amarasinghe, and K. Meyer, submitted to AMT*

*This manuscript describes a long overdue attempt at cloud optical thickness (COT) and effective radius (CER) retrievals from the ASTER instrument using a MODIS-like look-up table approach. Some adjustments have to be made because of spectral response function differences but overall the approach is very similar. As ASTER and MODIS are located onboard EOS-Terra that makes pixel-level comparisons (as done in this paper) straightforward with new insights gleaned on resolution-dependent issues between the two instruments.*

*This is an excellent paper that is very straightforward in its presentation and messaging with two primary purposes: to describe the retrieval, and compare the ASTER results to MODIS Collection 6 retrievals. A couple of case studies are presented in detail with images shown and details of the scene described at length, then 48 scenes are shown for statistical comparisons between ASTER and MODIS. In my opinion, there is little to improve upon in this paper and I only have a few minor revisions to suggest and comments to make below.*

**What about the land challenge? Can this approach be extended to land retrievals without much development? Are there plans to do so, and if not, why not?**

As mentioned in lines 323-324, one of the inputs for the retrieval algorithm is the gap–filled MODIS surface albedo product (Moody et al., 2005, 2007, 2008). ASTER even samples observations around λ=0.66 μm, a channel MODIS usually employs for retrievals over land surfaces. This means that there are no technical limitations preventing a cloud property retrieval over land. ASTER also has a channel centered around λ=1.6 μm, which (in combination with the usual 2.1 μm band) is suggested in Platnick et al. (2001) and Moody et al. (2007) for retrievals over sea ice.

However, retrievals over land require a more thorough testing of the cloud mask, together with possible adjustments to individual cloud mask tests. As for MODIS, cloud property retrievals over land induce larger uncertainties in retrieved cloud products due to the larger uncertainties in the highly variable surface albedo. This is the reason for just choosing scenes over the ocean for this comparison study.

The MODIS cloud mask algorithm MOD35 can not be applied directly to

ASTER because the MODIS cloud mask algorithm uses many more channels than are available on ASTER, notably the 1.38μm channel and many IR channels as described in Ackerman et al (2006), many more than ASTER carries.

We mention the possible adjustments to the cloud masking scheme for scenes with different characteristics (such as cloud scenes over land) in Section 3.1:
"As demonstrated below, the quality of the cloud mask tests meet the purpose of this study and are believed to be more broadly appropriate for deep ocean scenes, in atmospheres with low aerosol turbidity, and outside of strong sun–glint and large $\theta_0$. However, it should be noted that further refinements of these thresholds are likely for investigations outside the scope of this study."

Extending our cloud masking and cloud property retrievals to land is certainly on our to-do list for future research.

***p. 3 around lines 65-75: there is at least one other attempt at an ASTER cloud mask that could be considered for citation (if deemed relevant): Hulley, G. C., and S. J. Hook (2008), A new methodology for cloud detection and classification with ASTER data, Geophys. Res. Lett., 35, L16812, doi:10.1029/2008GL034644.***

Thanks for this relevant reference, we included this study in the introduction:
"Hulley and Hook (2008) describe a number of spectral tests to distinguish cloudy observations from those over different surfaces and cloud shadows."

***Lines 101-102: do the authors think that the backward-viewing direction has some potential to improve upon the ASTER COT and CER retrievals? If so, what geophysical field(s) would this benefit the most?***

The backward-viewing 0.86 band is indeed interesting. However, three factors complicate the application of this band in the cloud property retrieval:
(i)     The viewing geometry for the backward-viewing 0.86 band is different from all other channels. For a perfectly plane-parallel cloud this can be handled by an adjustment to the LUT. For clouds with a complex 3D structure and varying cloud top heights, however, 3D radiative effects can impact the two channels used in the retrieval in different ways. The two bands might even sample different parts of the cloud.
(ii)    There is a time lag between the samples in forward and backward direction.
(iii)   The backward-viewing image is offset with respect to the nadir image,

which requires complicated cross-registering of all pixels. Because ASTER pixels are so small (15m and 30m for the VNIR and SWIR band, respectively), pixel co-location needs to be extremely precise and small differences would induce additional uncertainty in the retrieval.

As mentioned in the introduction, Seiz et al. (2006) and Genkova et al. (2007) uses the stereoscopic capabilities of ASTER for cloud top height analysis. There The retrieval here is similar to the cloud top height retrieval approach by the Multi-angle Imaging SpectroRadiometer (MISR) described in Moroney et al. (2002). The retrieval of cloud top heights from ASTER might benefit from the backward-viewing channel.

*Section 3.2.1 cloud top properties: First, it wasn't fully clear how the ASTER pixel level data was averaged up to the MODIS resolution. Second, are all ASTER retrievals done at the ASTER resolution then averaged to the MODIS resolution for comparison, or are the reflectances/radiances averaged then the retrieval is performed? It appears that the former was the case for most, if not all, of the paper but it should be made clearer or in a few places if made already. Third, are retrievals compared if ASTER is partly cloudy within a cloud-identified MODIS pixel?*

The retrieval of cloud top properties, as well as optical and microphysical parameters, is independent from the MODIS resolution and the ASTER cloud top property retrieval does not require averaging to the MODIS resolution. Therefore, everything stated before Section 5 (i.e., the cloud masking scheme, retrieval algorithm, …), is completely independent from the MODIS resolution. Only later in Section 5, do we aggregate the ASTER observations within the MODIS geometry to show the viability of the retrieval algorithm.

We looked at the introduction of Section 3.2 again and believe that the confusion possibly stems from this part:
"… eliminates uncertainties when comparing retrieval products between the different sensors. This allows for a comprehensive comparison between the MODIS and ASTER results without biases due to the applied set of equations."

Again, Section 3 only documents the retrieval setup and differences to the operational MODIS code. Everything is completely independent from the MODIS resolution and, as shown in Figures 7a-c and 8a-c, yields retrievals at the native ASTER resolution of 15m. As a result, we shortened the introduction to Section 3.2:
"The research-level ASTER retrieval setup uses the same algorithms as the operational MODIS C6 retrieval, which has been extensively tested and documented."

Moreover, the introduction to Section 5 (i.e., the comparison between the ASTER and MODIS retrievals) now includes the following sentences:

"By employing the operational MODIS C6 retrieval algorithms, uncertainties in the comparison of retrieved cloud products from both sensors are mitigated. This allows for a comprehensive comparison between the MODIS and ASTER results without biases due to the applied set of equations."

We believe that these steps help in clearing things up, especially since information about aggregation immediately follow in the first sentence in Section 5.1, "... the high–resolution ASTER digital counts $d_A(\Delta\lambda)$ are aggregated within each $(1000\cdot1000)$ m MODIS pixel."

Regarding the retrieval for partially cloudy pixels: A retrieval is always attempted (if the pixel is considered cloudy), and in the comparison we differentiate between overcast and partially cloudy pixels. This fact is stated at the beginning of each individual subsection in the Section 5 (e.g., in lines 514, 548, 619 and 657).

**Line 413: constrained**

We fixed the spelling error.

**Lines 580-587: how do the ASTER uncertainties fall within the MODIS uncertainties? Does this work yield further insight on the character of MODIS COT and CER pixel-level uncertainty estimates?**

We now provide information on the derivation of pixel-level ASTER uncertainty estimates in Section 3.2:
"The current MODIS retrieval products provide pixel-level uncertainty estimates for $\tau$ and $r_{\text{eff}}$. Because the ASTER retrieval algorithm deploys the same retrieval code, ASTER pixel-level retrieval uncertainties are derived in a similar way. They are comprised of uncertainties in the applied surface albedo (15%), calibration and model uncertainties (5%) and uncertainties in the amount of above-cloud precipitable water, which are an input variable in the atmospheric correction (20%)."

This means that ASTER retrieval uncertainties are almost identical to the MODIS results, with two exceptions:
(i)      The gap-filled albedo values, employed for retrievals over land, are provided for the MODIS SRF.  (This is not important for this study, as we only compare maritime cloud scenes.)
(ii)      MODIS C6 pixel-level uncertainty estimates include scene-dependent L1B uncertainties, while ASTER uses a constant value.

Both differences to the operational MODIS C6 product are now mentioned in Section 5.4:
"Since the land surface albedo product is created for MODIS bands 1-7 and there is no specific surface albedo product for ASTER, the SRF differences

between ASTER and MODIS bands induce uncertainties in the derived spectral surface albedo values. This is acknowledged by an increase in surface albedo uncertainty from 15% to 30% in the pixel-level uncertainty calculations. However, since the focus of this study is on MBL clouds sampled over ocean, this effect is mitigated by the use of ocean surface reflectances derived from the Cox-Munk model generated using the precise ASTER SRF."

And:
"The pixel-level uncertainty estimates for retrieved cloud products based on ASTER observations are comprised of uncertainties in surface albedo, radiometric calibration, the applied models and the amount of above-cloud precipitable water. This closely follows the MODIS Collection 6 approach and yields similar retrieval uncertainties for both the ASTER and MODIS results. However, while for ASTER the calibration and model uncertainties are assumed to be a constant value of 5%, uncertainties in the operational MODIS C6 cloud products include spectral, scene-dependent Level-1B uncertainty indices (Sun et al., 2012)."

New insights into the MODIS pixel-level retrieval uncertainty estimates can be gathered by the high-resolution sub-pixel reflectance distribution. Following the discussion in Platnick et al. (2004) and Zhang et al. (2016), high-resolution reflectance samples can be used to explain, and possibly correct for, biases in retrieved $\tau$ and $r_{\text{eff}}$.

We added these outlooks in the Summary:
"ASTER observations at native resolution can help in determining the subpixel cloud structure of heterogeneous cloud pixels, which result in significant uncertainties in the cloud property retrieval (Marshak et al., 2006). Studies by Platnick et al. (2004) and Zhang et al. (2016) show that information about the subpixel reflectance distribution can be used to explain, and possibly correct for, biases in retrieved cloud optical thickness and effective droplet radius. Similar analysis on the distributions of subpixel $\gamma_{0.86,\,A}$, $\gamma_{2.1,\,A}$, $\tau_A$ and $r_{\text{eff},\,A}$ can improve the understanding of MODIS PCL retrievals and their uncertainties."

***Lines 753-756: it is somewhat unclear in the way currently described how the ASTER uncertainties are obtained. Maybe a graphic, table, or improved discussion will help clarify.***

We agree with the reviewer and added the following information in Section 3.2:
"The current MODIS retrieval products provide pixel-level uncertainty estimates for $\tau$ and $r_{\text{eff}}$. Because the ASTER retrieval algorithm deploys the same retrieval code, ASTER pixel-level retrieval uncertainties are derived in a similar way. They are comprised of uncertainties in the applied surface albedo (15%), calibration and model uncertainties (5%) and uncertainties in

the amount of above-cloud precipitable water, which are an input variable in the atmospheric correction (20%)."

Additional information about the MODIS and therefore ASTER pixel-level uncertainly calculations can be found in Platnick et al. (2016).

***Figure 3: the authors may want to consider labeling flag=0,1,2,3 also as the four categories of clear/cloudy and how they map to the MODIS cloud mask. Labeling as flag=0,1,2,3 makes the figure less useful because you have to page to the discussion to figure it out.***

> We agree with the reviewer and added the four categories to flags 0-3 in Figure 3. We also added a sentence in the Figure caption:
> "The results of tests (i)-(iv) yield a designation of cloudiness flag '0' (confidently cloudy), '1' (probably cloudy), '2' (probably clear) or '3' (confidently clear) for each pixel".
>
> The naming convention follows the MODIS designation, which is made clearer in Section 3.1, where a new sentence states:
> "Following the MODIS cloud mask designation presented in Platnick (2003), ASTER pixels can be flagged as confidently cloudy, probably cloudy, probably clear, or confidently clear."

***Figure 9: are MODIS pixels really perfect squares? How does the spatial weighting of the reflectance as seen by the instrument look like within the pixel? Same point about ASTER. Any references of previous work discussing current state of knowledge of pixel-level characteristics would benefit the methodology of the paper.***

> The reviewer's concerns about the shape of a MODIS pixel are valid, as MODIS pixels are not always perfect squares. We added the following information in Section 5.1:
> "It must be noted that while MODIS pixels from scene C19 can almost be considered squares, this is not universally true for all MODIS pixels. Scenes closer to the edge of a MODIS swath are characterized by an increase in pixel size along the scan direction."
>
> The reviewer is also correct that MODIS samples are characterized by a point spread function (PSF), where the largest contribution of a MODIS signal comes from the center of the pixel and slightly extents past the pixel borders. We mention this issue in Section 5.4, where we say:
> "Moreover, uncertainties arise due to the MODIS point spread function (PSF), which characterizes the signal distribution within and outside a MODIS pixel (Huang et al., 2002). While $d_A(\Delta\lambda)$ from all ASTER samples contribute equally to the aggregated signal, the MODIS PSF implies that the largest

contribution in a MODIS signal comes from the center of the pixel, while there is also a noticeable influence from surrounding pixels."

If we were interested in a thorough instrument characterization, or even cross-calibration, this would be an issue. Ideally, a similar PSF could be assumed for each 15m and 30m ASTER pixel. Subsequently, for each MODIS pixel, some sort of aggregated signal from each of these ASTER pixels (including their PSF) could be constructed, following the overall MODIS PSF. We have not done such an analysis. However, the goal of this paper is to show the feasibility of reliable, high-resolution ASTER retrievals. Because of the very good agreement, we feel comfortable in showing the retrieval comparison as presented in the manuscript, with the acknowledgment that the PSF induces uncertainties in the comparison.

***Figure 11: Impossible to see 'partial' in the upper row. May consider shrinking y-axis scale or presenting data in a different manner if seeing gray points is a key take-away from this figure.***

The upper row shows retrieval results for the very homogeneous scene C14. This case includes no partially cloudy pixels. As noted in the reply to reviewer 2, Figures 10 and 11 in the old manuscript were actually older versions of the correct Figures. We tested the effects of different cloud masking algorithms and LUT resolutions on the retrieval comparison and just implemented the wrong version of these Figures. We updated both Figure 10 and 11 to the final versions, using the correct cloud mask. We also removed the "Overcast" and "Partial" for the homogeneous case C14 to remove any possibility of confusion (as there are no partially cloudy pixels for this case).

Not affected were the statistical analysis plots Figure 12-14, which used the correct LUTs, as well as the correct version of the cloud masking algorithm.

We decided against shrinking the y-axis to put emphasis on the significant differences that are obvious for the inhomogeneous cloud scene shown in c) and d). We think that this also illustrates why the large differences between MODIS and ASTER retrievals, shown in Figures 11c-d for the inhomogeneous scene C19, are not observed for C14.

We apologize for the confusion and thank both reviewers for noticing that there was something wrong with the Figures.

**References in this response:**

Ackerman, A., K. Strabala, P. Menzel, R. Frey, C. Moeller, L. Gumley, B. Baum, S. W. Seemann, and H. Zhang, 2006: Discriminating clear-sky from cloud with MODIS Algorithm Theoretical Basis Document (MOD35). ATBD Reference Number: ATBD-MOD-06. modis-atmos.gsfc.nasa.gov/reference_atbd.php. LAD:09.13.2016

Genkova, I., Seiz, G., Zuidema, P., Zhao, G., and DiGirolamo, L.: Cloud top height comparisons from ASTER, MISR, and MODIS for trade wind cumuli, Remote Sens. Environ., 107, 211–222, doi:10.1016/j.rse.2006.07.021, 2007.

Marshak, A., Platnick, S., Varnai, T., Wen, G. Y., and Cahalan, R. F.: Impact of three-dimensional radiative effects on satellite retrievals of cloud droplet sizes, J. Geophys. Res., 111, 2006.

Moody, E. G., King, M. D., Platnick, S., Schaaf, C. B., and Gao, F.: Spatially complete global spectral surface albedos: Value–added datasets derived from terra MODIS land products, Ieee Transactions On Geoscience and Remote Sensing, 43, 144–158, doi:10.1109/TGRS.2004.838359, 2005.

Moody, E. G., King, M. D., Schaaf, C. B., Hall, D. K., and Platnick, S.: Northern Hemisphere five-year average (2000-2004) spectral albedos of surfaces in the presence of snow: Statistics computed from Terra MODIS land products, Rem, 111, 337—345, doi:10.1016/j.rse.2007.03.026, 2007.

Moody, E. G., King, M. D., Schaaf, C. B., and Platnick, S.: MODIS–Derived Spatially Complete Surface Albedo Products: Spatial and Temporal Pixel Distribution and Zonal Averages, J. Appl. Meteorol., 47, 2879–2894, doi:10.1175/2008JAMC1795.1, http://dx.doi.org/10.1175/2008JAMC1795.1, 2008.

Moroney, C., Davies, R., & Muller, J. -P.: Operational retrieval of cloud top heights using MISR data, IEEE Transactions on Geoscience and Remote Sensing, 40, 1532–1540, 2002.

Platnick, S., M. D. King, H. Gerber, P. V. Hobbs: A solar reflectance technique for cloud retrievals over snow and ice surfaces, J. Geophys. Res., 106, 15185-15199, 2001.

Platnick, S., R. Pincus, B. Wind, M. D. King, M. Gray, and P. Hubanks, 2004: An initial analysis of the pixel-level uncertainties in global MODIS cloud optical thickness and effective particle size retrievals. Passive Optical Remote Sensing of the Atmosphere and Clouds IV, S.-C. Tsay, T. Yokota, and M.-H. Ahn, Eds., *Proc. of SPIE*, **5652**, 30-40.

Platnick, S. , K.G. Meyer, M.D. King, G. Wind, N. Amarasinghe, B. Marchant, G.T. Arnold, Z. Zhang, P.A. Hubanks, R.E. Holz, P. Yang, W.L. Ridgway and J. Riedi, 2016: The MODIS cloud optical and microphysical products: Collection 6 updates and

examples from Terra and Aqua. IEEE T. Geosci. Remote Sens. TGRS-2016-00320, accepted.

Seiz, G., Davies, R., & Gruen, A. (2006). Stereo cloud-top height retrieval with ASTER and MISR. International Journal of Remote Sensing, 27(9) (May 10), 1839–1853.

Sun, J., Xiong, X., Madhavan, S., and Wenny, B. N.: Terra MODIS Band 27 Electronic Crosstalk Effect and Its Removal, IEEE T. Geosci. Remote Sens., 52, 1551–1561, 2014.

Zhang, Z., Werner, F., Cho, H.-M., Wind, G., Platnick, S., Ackerman, A. S., Di Girolamo, L., Marshak, A., and Meyer, K.: A framework for quantifying the impacts of sub-pixel reflectance variance and covariance on cloud optical thickness and effective radius retrievals based on the bi-spectral method, J. Geophys. Res. - in review, 2016.

---

## Author Comment (AC2) · 1 Nov 2016

* Overall comment This paper is a technical report that discusses retrievals of cloud properties obtained from ASTER (new generating) and MODIS C6 (existing). The significance of this paper is that this paper reveals bias and differences between ASTER and MODIS products. However, discussion is too long comparing with the volume of contents and a little bit dull. I recommend to authors to refine this manuscript e.g. shorten the paragraphs and focus on a few topics that authors like to emphasis the most.

Questions and minor correction

***Line 74: The sentence "there remains no cloud property retrieval algorithm for ASTR" contradicts to the sentence "(Line 66) Zhao and Di and Dey et al. use the high –resolution ASTER reflectance measurements at the lambda=0.86um band to derive a statistical description of the macrophysical proper- ties of trade wind clouds". Please clarify this point.***

> The referenced paper describes macrophysical properties, i.e., cloud extent and scene cloud cover.
>
> We rephrased this part as follows:
> "Despite all these studies, there remains no retrieval algorithm that provides cloud top, optical and microphysical properties for the high-resolution ASTER observations."

***Line 106 and Line 485, Line 723 etc.: I don't understand the meaning of symbol "delta" in dA(delta lambda).***

> The symbol $\Delta\lambda$ indicates that the radiance/reflectance is sampled within a specific wavelength range (i.e., a spectral band) and is not a monochromatic (or even close to monochromatic) variable.
>
> We explain this symbol in the following way:
> "… the respective wavelength ranges $\Delta\lambda$ …" in Line 100 (in the original submission)
>
> and:
>
> "… wavelength range ($\Delta\lambda$)…" in the caption of Table 1.

***Line 180: Is it reasonable to use the thresholds of 0.03 and 0.065 that obtained
from MODIS algorithm to ASTER analysis?***

The reviewer is correct, in that it is not reasonable to just adopt threshold
derived for another instrument. However, as stated in the manuscript the
cloud masking scheme is independent from the MODIS algorithm and is
performed with an ASTER-specific algorithm. The respective thresholds have
been derived from extensive analysis of over 210 MBL cloud scenes and just
happen to be similar to the MODIS thresholds, which is not so surprising
because similar spectral bands are used for cloud masking.

To make sure that we do not just adopt these thresholds we state in the
manuscript:
"Similar tests to identify clear–sky pixels have been reported by..."

and

"These thresholds, which comprise the first step in the new cloud masking
scheme, were set through inspection of 210 ASTER MBL scenes sampled..."

As shown in Section 3.3, this similarity is understandable, since the VNIR
bands of both sensors are very similar.

***In Fig 3 : 0.80>r1<1.75 seems an error. Please correct it.***

We fixed this error.

***Line 204: Is it reasonable to use the thresholds of 0.065, 0.02, 0.80, 1.75 that
obtained from MODIS algorithm to ASTER analysis?***

Again, this is just a coincidence. Following our reply to the similar question
earlier, we agree that simply adopting these thresholds from MODIS for
ASTER is not reasonable. However, for ASTER observations we apply an
ASTER-specific cloud masking scheme and thresholds have been developed
independently for over 210 MBL scenes. These thresholds happen to agree
with the MODIS thresholds, which is not surprising given the overall
agreement of both VNIR bands.

***Line 215: How to obtain the values of 33.4egs, 63.2degs, and aerosol optical
thickness of (0.04-1.49) ? Please clarify this point.***

They are the observed ranges of the 210 MBL scenes that were inspected to
derive the cloud masking thresholds.

We added this information to the manuscript:

"(observed \theta_0 = 33.4° - 63.2° for the 210 MBL scenes)" and "(observed 0.04-1.49 for the 210 MBL scenes)"

**Lin 230: How to scale up the TB11? Please clarify this point.**

We removed the reference to "scale up" and this part now just reads:
"In order to match the spatial resolution of the VNIR observations, each $T_{B,11}$ sample at 90m resolution is replicated onto 36 subpixels with a horizontal resolution of 15m."

This replication is also illustrated in the Figure below:

[Figure]

Figure 1: Sketch of replicating a single 90m $T_{B,11}$ sample at 90m resolution to match the ASTER VNIR resolution of 15m.

**Line 244: What does "a single case-by-case threshold for gamma0.86.A" mean?**

The cloud masking scheme introduced in Zhao and Di Girolamo (2006), Uses a single threshold for the sampled digital counts in the ASTER VNIR band (instead of multiple tests), which is determined individually on a case-by-case basis.
As an example, this threshold might be 120 (digital counts out of 255) for one scene, but 90 for another, and 101 for a third scene.
Please note that in the originally submitted paper we mistakenly wrote that Zhao and Di Girolamo (2006) apply a threshold for VNIR reflectances $\gamma_{0.86, A}$. This has been corrected in the revised manuscript.

**Line 255: -0.04> delta CA < 0.04 seems an error. Please correct it.**

We fixed this error.

**Line 296: Why do you use such a complex method such as "If the calculated cloud top pressure is larger than 650mb the operational MODIS C6 IR window retrieval algorithm is used to calculate the final value of cloud top pressure" ? Please clarify this point.**

As mentioned throughout the manuscript, we adopt the operational MODIS C6 retrieval algorithm for ASTER. MODIS applies the IR window technique when the $CO_2$-slicing algorithm, if the latter is unable to retrieve a valid cloud-top pressure. This can happen if the cloud signal signal in the thermal bands between 13.3 and 14.2µm is too low. More information on the applied IR window technique in C6 is provided by Baum (2012), which is also referenced in the manuscript.

***Line 337 and Line 326: There are two similar sentences to explain the Cox and Munk (1954a, b). Could you unify these two sentences into one?***

We changed the first mention of the Cox-Munk parameterization as follows: "This wind speed is used as input in the parameterization following Cox and Munk (1954a, b), which yields the wind speed-dependent bidirectional ocean surface reflectance."
We changed the second mention as follows:
"Similar to the correction of surface contributions, the ocean surface reflectance is obtained from the Cox-Munk parameterization, as implemented in the radiative transfer library libRadtran (Mayer and Kylling, 2005; Mayer, 2009)."

This changes help in avoiding the multiple explanations of the wind dependence and the double citation.

***Line 348: How to scale up SWIR (30m) in to 15m spatial resolution?***

We removed the confusing mention of "scale up" and the revised sentence reads:
"It must be noted that for the cloud property retrieval at 15m horizontal resolution each SWIR reflectance sample at 30m resolution is replicated onto 4 subpixels to match the band 3N resolution."

See also Figure 1.

***Line 370- and Eq (4): I cannot understand significance of examining the scale factor f0.86m LUT, because you have set up ASTER LUT and MODIS LUT, individually. There are no mean to examine ratio (scale factor) of LUT values. Please clarify this point.***

The reviewer is correct in that the difference between the respective ASTER and MODIS bands is accounted for in the respective LUTs. However, the scale factor is introduced to better compare sampled reflectances in Section 5.2.1 and 5.3.2.

We clarified this fact in the revised manuscript, by stating at the end of Section 3.3:
"The calculation of the ratios f0.86, L and f2.1, L allows for a direct comparison of measured ASTER and MODIS reflectances, as illustrated in Section 5.2.1. and Section 5.3.2."

**Line 406: It is difficult to understand "124 ASTER scenes" and "48 MBL cloud case".**

We thank the reviewer for noticing this artifact of an old version of the manuscript. In this older version we put more emphasis on distinguishing the different regions were the ASTER scenes were sampled. Overall, we currently have 210 ASTER MBL scenes, most of them sampled either off the Coast of California or during in the tropical Atlantic. These latter cases are trade wind cumulus scenes with a very low cloud cover. As a result, we limited the data set to scenes sampled off the Coast of California.

We rewrote this part as follows:

"For this reason, not all of the 210 ASTER MBL scenes, that were used in evaluating the 15m cloud mask in Section 3.1, are sufficient. The scenes sampled in the tropical western Atlantic (Zhao and DiGirolamo, 2006) were populated entirely by trade wind cumuli with a peak in the cloud fraction distribution at 400-500m in cloud equivalent diameters (see Zhao and Di Girolamo, 2007). The data set used in the following comparison consists of 48 MBL scenes sampled over the Pacific Ocean off the Coast of California between 05/2003 and 07/2007. Granules were manually chosen to include MBL clouds that resemble altocumulus or broken cumulus scenes and thus are characterized by extensive MBL cloud cover and cloud sizes."

We also changed "cloud cases" to "scenes"

**Line 412: 48 MBL cloud cases were sampled between 05/2003 and 07/2007. However, in line 123, it is mentioned that "the SWIR signal started to suffer from anomalous striping and saturation of values due to an increase in the SWIR detector starting in May 2007". Is it unreasonable to use 05/2007 to 07/2007 data for the cloud retrieval?**

The ASTER SWIR data issue is connected to the detector temperature. As noted by the ASTER team: "As long as the detector temperature remains at 83ºK, little or no degradation of ASTER SWIR data is expected." Information and graphs on this issue can be found on the ASTER science team website: https://asterweb.jpl.nasa.gov/swir-alert.asp

For a brief period in June and July 2007 attempts to reduce the detector temperature were successful and the SWIR data quality was fine. As mentioned in the manuscript, we made sure that the scenes from 2007 were not impacted by the temperature problems of the SWIR detector.

**Line 416: How to eliminate overlying cirrus scenes and complex multi-layered cloud system?**

From the whole data set we selected possibly suitable scenes based on the inspection of the RGB image of each scene. Subsequently, the MODIS cloud phase and cloud top height retrievals were used to eliminate scenes with thin cirrus clouds that were not visible in the RGB maps.

**Line 457: Why there appeared cloud holes in MODIS than ASTER?**

The MODIS retrieval is performed at 1km, while the ASTER retrieval is performed at 15m. The lower resolution of MODIS means that the pixel values are aggregated over low cloud reflectances and even lower ocean surface reflectances. This means, that the 1km pixel values is too low and the retrieval fails, where ASTER yields successful retrievals for the low cloud reflectances.

We added this additional information to the manuscript:
"…there are visibly more MODIS pixels throughout the granule where the retrieval fails. These pixels are characterized by $R_{0.86,M}$ and $R_{2.1,M}$ (sampled at 1000 m) that are too low for a successful cloud property retrieval (i.e., measurements fall outside the LUT)."

**Line 468: It is difficult to identify "visible striping in the reff, M results in this figure.**

The striping is definitely not as pronounced for the presented scenes, but it can be much worse for granules. However, people familiar with MODIS measurements noticed the striping in these scenes. For that reason we decided to include this sentence in order to avoid confusion with some readers.

In the Figure below (a close-up of Figure 8d) slight striping becomes apparent.

[Figure]

Figure 2: Slight striping visible in a close-up of Figure 8d.

***Line 497: what does 4400 and 1100 pixels mean in this sentence.***

A MODIS pixel has a horizontal resolution of 1000x1000m. VNIR and SWIR ASTER pixels are characterized by 15x15m and 30x30m resolutions, respectively. This means that a MODIS pixel contains (1000/15 * 1000/15) > 4400 ASTER VNIR pixels.

We added the following information in the revised manuscript:
"Taking into account the different spatial resolutions of both instruments (1000m for MODIS, 15m for the ASTER VNIR band, 30m for the ASTER SWIR band)…"

***Line 549: It is difficult to identify black circles and gray circles in the Fig.11 because the figure size is too small. It may be a technical problem but significant to audiences. Please improve Fig. 11.***

The upper row shows retrieval results for the very homogeneous scene C14. This case includes no partially cloudy pixels. As noted in the reply to reviewer 1, Figures 10 and 11 in the old manuscript were actually older versions of the correct Figures. We tested the effects of different cloud masking algorithms and LUT resolutions on the retrieval comparison and just implemented the wrong version of these Figures. We updated both Figure 10 and 11 to the final versions, using the correct cloud mask. We also removed the "Overcast" and "Partial" for the homogeneous case C14 to remove any possibility of confusion (as there are no partially cloudy pixels for this case).

Not affected were the statistical analysis plots Figure 12-14, which used the correct LUTs, as well as the correct version of the cloud masking algorithm.

We decided against shrinking the y-axis to put emphasis on the significant differences that are obvious for the inhomogeneous cloud scene shown in c) and d). We think that this also illustrates why the large differences between MODIS and ASTER retrievals, shown in Figures 11c-d for the inhomogeneous scene C19, are not observed for C14.

We apologize for the confusion and thank both reviewers for noticing that there was something wrong with the Figures.

*Line 573: I cannot find "-12.36um" of r_eff, AaM- r_eff,M in the figure 11.*

We thank the reviewer for noticing this error. As noted in the previous reply, this was an older version of the Figure, where we tested the effects of different cloud mask algorithms on the retrieval comparison. We updated the Figure to the final version, using the correct cloud mask. This error also affected Figure 10, which has likewise been corrected. Not affected were the statistical analysis plots Figure 12-14, which used the correct version of the cloud masking algorithm.

*Line 663: What does "Tau_M, Tau_AaM=0.17-24.92." mean? I cannot find that range of values of Tau between 0.17 to 24.92 in Fig. 14(b).*

The lowest retrieved optical thickness from the 48 MBL scenes is 0.17. These numbers are mentioned to give the reader the observed ranges (similar to the mention of the observed ranges in the reflectance comparison). The (0-1)-bin in Figure 14b shows such low optical thickness observations for both ASTER and MODIS. For the very large optical thicknesses there are only very few observations for PCL pixels, which makes the PDF value very low. We feel that logarithmic axis would not help the visibility of Figures 14a-14d.

*Line 668: What does "r_eff,M, r_eff, AaM = 4.76um"? I cannot find that 4.76um will have a specific values in Fig. 14 (c ).*

The lowest retrieved effective droplet radius from the 48 MBL scenes is 4.76μm. As before, we wanted to mention the observed value range of retrieval results for the comparison (similar to the mention of the observed ranges in the reflectance comparison). As seen in Figure 14c there are some observations in the bin 4-5μm, which includes this specific value.

*Line 672: 10.00um>r_eff,M, r_eff,AaM<20.00um will be an error.*

We fixed this error.

*Line 679: 0>Tau_AaM, Tau_M< 5 will be an error.*

We fixed this error.

*Line 710: Where does "15% to 30%" come from. Can you add some references?*

The MODIS MCD43D product yields daily 16-day (averaged) BRDF/Albedo data. Information about this product and its uncertainties can be found online at:
https://lpdaac.usgs.gov/dataset_discovery/modis/modis_products_table/mcd43d61_v006
or in, e.g., Mira (2015).

The 15% albedo bias can be attributed to measurement uncertainties and the fact that the 16 day average is not the actual value at the time of the cloud retrieval. Information about the albedo uncertainty is reported in the Summary of the Changes in the MODIS Collection 5 Cloud Optical Property Retrieval Algorithm. It can be accessed online at: http://modis-atmos.gsfc.nasa.gov/_docs/C005CloudOpticalPropertiesver3.3.pdf

In the ASTER retrieval algorithm we use the MODIS BRDF/Albedo CMG Gap-Filled Snow-Free Product MCD43GF V005, which is reported for the MODIS spectral bands. There is no albedo product for ASTER, so we have to make due with the MODIS results.

The 30% value is our estimation of the uncertainty for ASTER, due to the fact that we use the MODIS surface albedo product. It is very conservative and for measurements over the ocean probably too high. As mentioned throughout the manuscript, the algorithm is a research-level retrieval algorithm and for now this value is our best guess for the surface uncertainty contribution.

***Line 749: I cannot identify visible striping in Fig 8(b) and 8(d).***

Please see our answers to the similar question: "***Line 468: It is difficult to identify "visible striping in the reff, M results in this figure. "***

**References:**

Baum, B. A., Menzel, W. P., Frey, R. A., Tobin, D., Holz, R. E., Ackerman, S. A., Heidinger, A. K., and Yang, P.: MODIS Cloud–Top Property Refinements for Collection 6, J. Appl. Meteor. Climatol., 51, 1145–1163, doi:10.1175/JAMC-D-11-0203.1, 2012.

Maria Mira, Marie Weiss, Frédéric Baret, Dominique Courault, Olivier Hagolle, Belén Gallego-Elvira, Albert Olioso, The MODIS (collection V006) BRDF/albedo product MCD43D: Temporal course evaluated over agricultural landscape, Remote Sensing of Environment, Volume 170, 1 December 2015, Pages 216-228, ISSN 0034-4257, http://dx.doi.org/10.1016/j.rse.2015.09.021.

Zhao, G. and Di Girolamo, L.: Cloud fraction errors for trade wind cumuli from EOS-Terra instruments., Geophys. Res. Lett., 33, L20 802, doi:10.1029/2006GL027088, 2006.

Zhao, G. and Di Girolamo, L.: Statistics on the macrophysical properties of trade wind cumuli over the tropical western Atlantic, J. Geophys. Res., 112, 2007.

---

## Author Comment (AC3) · 1 Nov 2016

We'd like to thank the editor for handling our manuscript, as well as reviewer #3 for reading our manuscript and providing numerous, helpful comments. We have carefully read through all the comments and questions and revised the manuscript accordingly. Please find our point-to-point response to reviewer #3 below. Here, the reviewer's general remarks are formatted to be left-aligned text in italic font, the specific questions/comments are shown in left-aligned text in bold and italic font, while our responses are indented and formatted in regular font.

**Point-to-point Response to Anonymous Referee #3**

**General**

The paper addresses an important research question on the applicability / validity of 1D radiative transfer calculations for high spatial resolution cloud property retrievals. Thereto the authors adapt an existing retrieval scheme for large pixels from MODIS to small pixels from ASTER.
This paper fits well into AMT. The scale dependence of cloud property retrievals is very important question and the research is very relevant for interpretation of satellite cloud retrievals.
The paper is well-written, contains important results based on solid work, and will probably lead to future papers on the same topic. The number of figures is large, and could possibly be reduced. The paper can be accepted when the following comments are taken into account.

**Main comments:**

*(1) The complication of atmospheric absorption in the wide VNIR band of ASTER (channel 3) is hardly discussed. Atmospheric correction has a much stronger effect for ASTER due to its broad VNIR band than for MODIS. In the broad VNIR channel of ASTER, the O2 A-band absorption and H2O band absorption play a large role. Please describe how you correct for these atmospheric absorption bands. The correction will depend on cloud height: the lower the cloud, the more correction is needed. Please show the sensitivity of the correction to cloud height. Please show the atmospheric absorption spectrum together with the spectral response function of the instrument bands in Figure 1.*

The reviewer is correct, in that the ASTER VNIR band includes the alpha band of atmospheric oxygen, as well as absorption features by atmospheric water vapor. This means that atmospheric correction (and uncertainties associated with atmospheric correction) will be more important for ASTER than for MODIS. Figure 1 of this reply shows the ratio of atmospherically corrected to uncorrected (measured at top-of-atmosphere, TOA) reflectance in the (a) VNIR and (b) SWIR bands as a function of cloud top height $z_{top}$. ASTER data is shown in black; MODIS data is shown in green.

As expected, the ratio (both in the VNIR and SWIR) approaches unity for large $z_{top}$, whereas the atmospherically corrected reflectances increase (compared to TOA reflectances) with decreasing $z_{top}$.

[Figure]

Fig. 1.: (a) Ratio $R_{cor}/R_{toa}$ as a function of $z_{top}$ for the VNIR band of both ASTER and MODIS. $z_{top}$ has been assigned to the y-axis. (b) Same as (a) but for the ASTER and MODIS SWIR bands.

We calculated the sensitivity $S$ of this correction as follows:

$$S = \frac{d(R_{cor}/R_{TOA})}{dz_{top}} \cdot \frac{z_{top}}{R_{cor}/R_{TOA}} = \frac{d\ln(R_{cor}/R_{TOA})}{d\ln(z_{top})} ,$$

where $R_{cor}$ indicates the atmospherically corrected and $R_{toa}$ is the uncorrected (measured at TOA) reflectance. This definition of sensitivity follows the relative sensitivity (susceptibility) definition used in e.g., Feingold et al. (2003), Werner et al. (2014) and others.

For the SWIR band both ASTER and MODIS are characterized by very similar sensitivities of -0.025 and -0.024. The negative sign indicates the decrease of the ratio $R_{cor}/R_{toa}$ with increasing $z_{top}$. For the VNIR band, conversely, $S$ is about four times higher for ASTER than for MODIS, with values of -0.021 and -0.006, respectively.

However, the applied atmospheric correction scheme uses the exact same ancillary data sets and algorithms as the operational MODIS C6 retrieval. Details of this algorithm are documented in Section 3.2.2 of the manuscript and the referenced literature. This means that the same well tested and successfully applied atmospheric correction scheme is used for both ASTER and MODIS in our comparison. While it is true that uncertainties associated with this scheme will induce larger uncertainties in the ASTER VNIR band (compared to MODIS), the very good comparison of ASTER and MODIS reflectances (see Figure 13) and subsequently retrieved cloud optical thicknesses (see Figure 14) give us confidence that the atmospheric correction scheme works reliably.

Based on the reviewer's suggestions, we made the following changes to the revised manuscript:

(i) We added atmospheric transmittance lines, based on atmospheric profiles for the U.S. 1976 Standard Atmosphere and simulations with the moderate resolution atmospheric transmission (MODTRAN) version 4.2r1. These curves show the absorption features of the $O_2$-Alpha band and atmospheric water vapor.

A brief description of these features is added to Section 2.3:
"The center position and width of the ASTER VNIR band implies that measurements are affected by important absorption features of atmospheric oxygen ($O_2$-A band around 0.760µm) and water vapor (mainly between 0.810-0.840µm). These features become apparent in the atmospheric transmittance spectrum $T_{atm}$ (grey), which was derived by simulations with the moderate resolution atmospheric transmission (MODTRAN) code version 4.2r1 (Berk et al., 1998), assuming profiles for atmospheric gases following the U.S. 1976 Standard Atmosphere. The atmospheric correction scheme, which accounts for these absorption features, as well as the associated uncertainty is described in Section 3.2 and 5.4."

(ii) We added the sensitivity discussion in the "Uncertainty Contribution" section (5.4):
"Because the ASTER VNIR band covers absorption features of atmospheric oxygen ($O_2$–A band) and water vapor, it is more sensitive to the atmospheric correction scheme than the respective MODIS VNIR band. The sensitivity has been derived by means of a susceptibility analysis, similar to the method described in Werner et al. (2014). The susceptibility $S$ is defined as the relative change of the ratio of uncorrected to corrected reflectance ($\hat{R}_{0.86,aA}/R_{0.86,aA}$) with a change in cloud top height $z_{top}$, which for the collocated ASTER VNIR data can be written as:

$$S = \frac{d(R_{cor}/R_{TOA})}{dz_{top}} \cdot \frac{z_{top}}{R_{cor}/R_{TOA}} = \frac{d\ln(R_{cor}/R_{TOA})}{d\ln(z_{top})} \quad . (5)$$

Deriving $S$ for all 48 MBL cloud scenes yields similar values of −0.025 and −0.024 for the ASTER and MODIS SWIR bands, respectively, indicating a similar sensitivity towards the atmospheric correction for both instruments. Conversely, $S$ in the VNIR bands is −0.021 (ASTER) and −0.006 (MODIS), indicating that measurements in the ASTER VNIR band

are significantly more sensitive to the atmospheric correction scheme than the respective MODIS measurements. This also implies that sampled reflectances in the ASTER VNIR band are more susceptible to uncertainties in the atmospheric correction scheme. However, the research–level retrieval algorithm presented in this manuscript employs the same ancillary data sets, as well as the extensively documented and tested atmospheric correction algorithm implemented in the operational MODIS C6 code. The good agreement between $\hat{R}_{0.86,aA}$ and $\hat{R}_{0.86,M}$, shown in Figure 13(e)–(f), can be attributed to the reliability of this scheme."

**(2) Can you already give conclusions on 3D effects seen in cloud retrievals from ASTER?**

Because the scope of this paper is to prove the documentation of the retrieval algorithm and the feasibility of cloud property retrievals with ASTER, the data shown in this manuscript do not provide the means to discuss impacts of 3D radiative effects. In fact, we aggregated the high-resolution ASTER observations within the MODIS resolution, thus intentionally reducing the native ASTER resolution to the same 1km. The only high-resolution retrieval results are shown exemplary in Figures 7-8, providing (naturally) much more detail than the MODIS retrievals.

However, based on this study and the documented retrieval algorithm, we are currently working on a study on the scale dependence of the plane-parallel bias, using high-resolution ASTER data. We are also working on a manuscript that uses the theoretical framework presented in Zhang et al. (2016) to correct for the plane-parallel bias based on subpixel reflectance variability. A third study concentrates on partially cloudy pixels and i) whether MODIS can reliably discriminate between overcast and PCL pixels, ii) some MODIS retrievals are biased because of a false overcast classification and iii) whether high-resolution retrievals over overcast and PCL pixels (the cloudy part) differ and how this changes with scale.

This first manuscript, which details the ASTER retrieval algorithm and proves the feasibility and reliability of the ASTER results, provides the technical basis for these future studies.

Minor and textual comments:

Abstract:

**l. 8, l. 10, etc.: symbols with subscripts lead to too heavy notation. Please shorten where possible.**

> The subscripts are necessary to distinguish between the LUT, MODIS, ASTER and aggregated ASTER variables without introducing new variable notations for each quantity. We carefully considered shortening all subscripts throughout the manuscript. While we decided to keep the wavelength designation (i.e., "0.65", "0.86", "2.1"), we shortened the subscript "AaM" (ASTER aggregated in MODIS) into "aA" and "LUT" into "L" (similar to "A" for ASTER and "M" for MODIS).

**\gamma is a strange symbol for reflectance. Please use R or \rho.**

> We used \gamma to follow the notation on page 159 in Wendisch and Yang (2012). However, since it is technically not the BRDF we are discussing and since in satellite remote sensing capital *R* is more widely used, we agree that \gamma is not the best symbol choice. We changed it throughout the manuscript.

**l. 13: There are too many details in the abstract.**

> We shortened and simplified the abstract by leaving out information about the subpixel cloud cover, scene statistics and the variable names for the atmospherically corrected reflectances.

**l. 20 ff: So is 1D retrieval good enough at these small scales? This is an important finding. Are there biases due to 3D RT for fully cloud covered pixels ? The effect of broken clouds on biases in reff is well known for 1D RT clouds. (E.g. Wolters et al., JGR, vol. 115, D10214, doi:10.1029/2009JD012205, 2010).**

> As mentioned in our reply to "Main comment #2", the statistical comparison presented in this manuscript is performed with co-located ASTER observations, which are derived from an aggregation of the high-resolution data within the MODIS geometry. This is done because we want to document the ASTER retrieval algorithm and prove the feasibility of a cloud property retrieval from ASTER observations. The only high-resolution retrieval results in the manuscript are shown exemplary in Figures 7-8. The only conclusions that can be drawn at this point, qualitatively, are i) that there is a good agreement in the patterns and absolute values of $\tau$ and $r_{\text{eff}}$ and that there is (naturally) a lot more detail in the high-resolution retrievals. We point these facts out in Sections 4.2 and 6.

**Section 1: At the end of the introduction, please briefly outline the setup of the paper.**

We included a brief outline at the end of Section 1:
"The manuscript is structured as follows: an overview of ASTER and MODIS, as well as difference between important spectral bands of the two instruments, is given in Section 2. The applied cloud masking scheme and the ASTER–specific cloud property retrieval algorithm are presented in Section 3. Subsequently, a comparison of the retrieval products between the operational MODIS C6 and collocated ASTER results is shown in Section 5, followed by summary in Section 6."

**l.101: in this spectral range oxygen and H2O absorption might be a problem**

The reviewer is correct, in that there are absorption features in the ASTER VNIR band, mainly caused by the alpha band of atmospheric oxygen, as well as atmospheric water vapor.

We detailed all the changes in the revised manuscript in our response to "Main comment #1".

**l. 118-120: These solar spectrum references are pretty old. Why not use a modern composite synthetic solar spectrum, like Gueymard (Solar Energy, 2004)?**

Indeed, the solar spectra used here are older than the Gueymard spectra. However, these are the solar irradiance values used in the current version of the MODIS retrieval algorithm. To avoid any bias in the comparison between ASTER and MODIS results, we chose to derive reflectances using the same input solar irradiance values. However, these values can be easily replaced by newer spectra in future applications, where a comparison with MODIS is not the focus of the study.

**l. 142: Please remove the brackets. This occurs at many places in paper.**

We changed it throughout the paper.

**l. 312: This correction will depend on cloud top height.**

The reviewer is correct, in that the atmospheric correction is dependent on the cloud top height retrieval. The applied MODIS C6 retrieval algorithms use cloud top height as an input for atmospheric correction. As mentioned in the summary, retrieved cloud top heights from MODIS and collocated ASTER retrievals agree well, with mean values of 823m (ASTER) and 670m (MODIS).

We included information about the sensitivity of the ASTER VNIR and SWIR signal to the atmospheric correction scheme in Section 5.4 and also added this small passage to the introduction of the atmospheric correction in Section 3.2.2:

"Atmospheric correction, which is a function of cloud top height, is performed by…"

**l. 386: what is the reason of this difference?**

We decided to rewrite this section of the paper for a couple of reasons. The most important one is that Figure 5 and the respective discussion did not sufficiently explain the impact of the different SRFs on the retrieval. In the originally submitted manuscript version we only showed a specific case (with a low solar zenith angle).

The revised version includes the following changes:
(i)     For two solar and viewing geometries complete ASTER and MODIS LUTs are presented, illustrating that the ASTER SWIR band is always brighter than the respective MODIS band. In contrast, the specific geometry determines, whether the ASTER VNIR band is slightly brighter or darker.
(ii)    We include details about the underlying physical explanations for the band differences between the two instruments. Specifically we state at the beginning of the Section:
"The discussion in Section 2.3 showed that there are differences between the VNIR and SWIR SRFs of ASTER and MODIS, which requires the calculation of ASTER-specific LUTs where the spectral scattering properties (i.e., extinction coefficient, single-scattering albedo and scattering phase function) are integrated over the ASTER SRFs."

and:

"The shift towards a larger center wavelength for the ASTER SWIR band yields an increase in scattering efficiency and single-scattering albedo. As a result the ASTER SWIR bands appears significantly brighter than the respective MODIS band …"

Regarding the reviewer's question: In the region of the two VNIR bands there is an increase in extinction efficiency $Q_e$ and a slight decrease in single-scattering albedo $\omega$ with increasing wavelength. Regarding the scattering efficiency, both tendencies basically offset each other. This is the reason for the good agreement between both sensors in the VNIR band (and subsequently $f_{0.86, L} \approx 1$), as well as the visibly white appearance of clouds.

The change in wavelength also affects the scattering phase function and this impact is different from scene to scene.
The impact of the SRFs on the scattering properties is clearer for the SWIR band. As mentioned in the revised manuscript, both $Q_e$ and $\omega$ increase with wavelength in the spectral region covered by both SWIR SRFs, leading to the brighter appearance of the ASTER SWIR band.

***l. 396-398: what is the physical reason that the ASTER observation is brighter?***

Please see out response to the earlier question ("***l. 386: what is the reason of this difference?***")

***l. 704-705: Absorption by O2 and H2O in the VNIR band does not only affect above- cloud correction, but also the cloud reflectance itself due to multiple scattering and absorption inside the clouds.***

The reviewer is correct. However, these effects are accounted for in the forward model used to generate the LUTs. As mentioned in the manuscript, after the above-cloud atmospheric correction a Rayleigh scattering correction is applied. Both steps are identical to the operational MODIS C6 retrieval.

***Figure 2: what kind of scene is this ? what about cloudiness ? please also give the gray scale image of the scene.***

This scene is from the RICO campaign (Rauber et al., 2007). It is comprised of a multitude of small, individual trade wind cumuli. Overall, the scene cloud cover is 4%. We added the following information to the manuscript:
"This scene is characterized by a multitude of individual cumuli with small horizontal extent and a low scene cloud cover of $C_A = 0.04$."

The gray scale image naturally looks very similar to Figure 2(a), i.e., the VNIR reflectances. This similarity, and the fact that Figure 2 already contains 6 subfigures (which would get smaller if we added the gray scale image), means that we decided against including the gray scale image in the revised manuscript. However, we included it in this response:

[Figure]

Fig. 2.: Single-band grayscale image of band 3N reflectances sampled by ASTER on 12/02/2004 in the tropical western Atlantic. More information on this and similar cases is provided in Zhao and Di Girolamo (2006) and Zhao and Di Girolamo (2007).

**Figure 5: Caption: is the solar azimuth also the viewing – solar azimuth difference, which is the relevant quantity? What is the scale factor?**

Please see out response to the earlier question ("***l. 386: what is the reason of this difference?***"). We now include model simulations for two different geometries. The relative azimuth angles are now stated for the two cases, both in text and in the Figure caption.

We referred to the variables $f_{0.86, L}$ and $f_{2.1, L}$ as scale factors because they resembled the theoretical, scene-dependent scale factor between ASTER and MODIS reflectances in the respective spectral bands. However, we simplified the description and now refer to the variables $f_{0.86, L}$ and $f_{2.1, L}$ simply as reflectance ratios.

**Figure 6: how large is this scene?**

ASTER scenes cover an area of 60x60 km. We mention this fact in Line 103.

**Figure 7: please refer to the previous figure.**

The captions of Figure 7 and 8 now both include the following sentence: "The corresponding single-band grayscale images of ASTER band 3N and MODIS band 2 reflectances are shown in Figure 6(a)-(d)."

**References:**

Berk, A., Bernstein, L. S., Anderson, G. P., Acharya, P. K., Robertson, D. C., Chetwynd, J. H., and Adler-Golden, S. M.: MODTRAN cloud and multiple scattering upgrades with application to AVIRIS, Remote Sens. Environ., 65, 367—-375, 1998.

Feingold, G., W. L. Eberhard, D. E. Veron, and M. Previdi (2003), First measurements of the Twomey indirect effect using ground-based remote sensors, Geophys. Res. Lett., 30(6), 1287, doi:10.1029/2002GL016633.

Robert M. Rauber, Harry T. Ochs III, L. Di Girolamo, S. Göke, E. Snodgrass, Bjorn Stevens, Charles Knight, J. B. Jensen, D. H. Lenschow, R. A. Rilling, D. C. Rogers, J. L. Stith, B. A. Albrecht, P. Zuidema, A. M. Blyth, C. W. Fairall, W. A. Brewer, S. Tucker, S. G. Lasher-Trapp, O. L. Mayol-Bracero, G. Vali, B. Geerts, J. R. Anderson, B. A. Baker, R. P. Lawson, A. R. Bandy, D. C. Thornton, E. Burnet, J-L. Brenguier, L. Gomes, P. R. A. Brown, P. Chuang, W. R. Cotton, H. Gerber, B. G. Heikes, J. G. Hudson, P. Kollias, S. K. Krueger, L. Nuijens, D. W. O'Sullivan, A. P. Siebesma, and C. H. Twohy, 2007: Rain in Shallow Cumulus Over the Ocean: The RICO Campaign. *Bull. Amer. Meteor. Soc.,* 88, 1912–1928, doi: 10.1175/BAMS-88-12-1912.

Wendisch, M. and P. Yang (2012), Theory of Atmospheric Radiative Transfer - A Comprehensive Introduction, Wiley-VCH Verlag GmbH \& Co. KGaA, ISBN: 978-3-527-40836-8

Werner, F., F. Ditas, H. Siebert, M. Simmel, B. Wehner, P. Pilewskie, T. Schmeissner, R. A. Shaw, S. Hartmann, H. Wex, G. C. Roberts, and M. Wendisch (2014), Twomey effect observed from collocated microphysical and remote sensing measurements over shallow cumulus, J. Geophys. Res. Atmos., 119, 1534–1545, doi: 10.1002/2013JD020131.

Zhang, Z., F. Werner, H.-M. Cho, G. Wind, S. Platnick, A. S. Ackerman, L. Di Girolamo, A. Marshak, and K. Meyer (2016), A framework based on 2-D Taylor expansion for quantifying the impacts of subpixel reflectance variance and covariance on cloud optical thickness and effective radius retrievals based on the bispectral method, J. Geophys. Res. Atmos., 121, doi:10.1002/ 2016JD024837.

Zhao, G. and Di Girolamo, L.: Cloud fraction errors for trade wind cumuli from EOS-Terra instruments., Geophys. Res. Lett., 33, L20 802, doi:10.1029/2006GL027088, 2006.

Zhao, G. and Di Girolamo, L.: Statistics on the macrophysical properties of trade wind cumuli over the tropical western Atlantic, J. Geophys. Res., 112, 2007.